# Transferrable Surrogates in Expressive Neural Architecture Search Spaces

Shiwen Qin[*1]  Gabriela Kadlecová[*2,3]  Martin Pilát[2]
Shay B. Cohen[1]  Roman Neruda[3]  Elliot J. Crowley[1]  Jovita Lukasik[†4]  Linus Ericsson[†1]

[1]University of Edinburgh
[2]Charles University, Faculty of Mathematics and Physics
[3]The Czech Academy of Sciences, Institute of Computer Science
[4]University of Siegen

**Abstract**  Neural architecture search (NAS) faces a challenge in balancing the exploration of expressive, broad search spaces that enable architectural innovation with the need for efficient evaluation of architectures to effectively search such spaces. We investigate surrogate model training for improving search in highly expressive NAS search spaces based on context-free grammars. We show that i) surrogate models trained either using zero-cost-proxy metrics and neural graph features (GRAF) or by fine-tuning an off-the-shelf LM have high predictive power for the performance of architectures both within and across datasets, ii) these surrogates can be used to filter out bad architectures when searching on novel datasets, thereby significantly speeding up search and achieving better final performances, and iii) the surrogates can be further used directly as the search objective for huge speed-ups.

## 1 Introduction

Neural architecture search (NAS) promises to find high performing architectures for diverse tasks, but the field has so far struggled to discover truly novel architectures. This challenge arises from the inherent trade-off between designing focused search spaces for specific architectural families, such as ConvNets (Dong and Yang, 2020), transformers (Chen et al., 2021a,b) and hybrids (Li et al., 2021; Thomas et al., 2025)—and the need for broader, more expressive search spaces that can enable true architectural innovation.

Recent work has proposed large and expressive search spaces based on context-free grammars (Schrodi et al., 2023; Ericsson et al., 2024). However, searching these spaces becomes more difficult as the size increases, meaning that efficient evaluation is more important that ever. Techniques such as performance predictors (White et al., 2021; Dudziak et al., 2020; Lukasik et al., 2024; Jawahar et al., 2024), surrogate models (Ning et al., 2021; Zela et al., 2020; Yan et al., 2021) and zero-cost proxies (Krishnakumar et al., 2022; Kadlecová et al., 2024; Mellor et al., 2021) offer a promising direction for this. While these techniques have shown impressive results on constrained search spaces, their performance in more expressive, complex spaces remains unknown. Many zero-cost proxies (ZCPs) rely on simple heuristics (e.g. Mellor et al. (2021) effectively counts convolutions) which may fail to capture the nuances of fundamentally different architectures like transformers.

In this work, we demonstrate that existing ZCPs struggle in expressive search spaces, but more recent methods that incorporate topology-based features yield better predictive performance. Additionally, we explore the capabilities of large language models (LLMs), which can interpret

---

* Equal contribution.
† Equal contribution.

structured, context-free grammar (CFG)-based representations of architectures. We show that these models can effectively predict architecture performance, significantly accelerating search across multiple datasets. Furthermore, while seeding search from known baseline architectures—such as ResNets—has proven useful (Ericsson et al., 2024), it introduces a bias toward specific architectural directions. Unseeded search has the potential to explore a broader solution space, yet improving search speed is necessary to make it feasible. Our surrogate models not only enhance search efficiency but also exhibit strong generalisation to unseen tasks, allowing them to function as global surrogates across diverse tasks. Finally, we explore the potential of using these surrogate models directly as search objectives, paving the way for more efficient and effective NAS. Our contributions are as follows:

- We evaluate a broad set of performance predictors in an expressive search space, and to our knowledge, we are the first to do so.

- We introduce a novel surrogate based on a fine-tuned LM which takes in string representations of the architecture derivations, achieving the highest correlations with real architecture performances.

- When using the surrogates during evolutionary search, we find that we can speed up the search significantly while consistently achieving stronger final architecture performances than baseline search without a surrogate.

- We can even use our surrogates directly as the search objective of NAS, offering huge speed improvements, and at times even beating the baseline that takes many times longer to run.

## 2  Related Work

### 2.1  Neural Architecture Search

Cell-based search spaces have been the dominant design type in NAS research. These search spaces are restricted to a single cell with only a few nodes and edges, and the architecture is created by repeatedly stacking the same cell. Examples include NAS-Bench-101 (Ying et al., 2019), NAS-Bench-201 (Dong and Yang, 2020), and DARTS (Liu et al., 2019a). The main advantage of this design is reduced search costs compared to searching in unrestricted spaces. However, this also means the design of architectures is limited, and truly novel architectures cannot be discovered.

Recently, focus has shifted to more flexible search-spaces with potential to discover novel architectures tailored to diverse datasets. These search spaces are defined by a *grammar* that enforces a specific structure. The first such search space, Hierarchical NAS (Schrodi et al., 2023), introduced a grammar for a flexible cell structure and macro architecture. The second was einspace—a search space with a flexible grammar focused on a basic set of operations (such as branching or routing). This flexibility enabled it to represent a wide range of architectures—convolutional networks, vision transformers and MLP mixers—as well as novel unexplored architectures (Ericsson et al., 2024).

### 2.2  Performance Predictors in NAS

Performance predictor models have been widely used in NAS to speed up the evaluation of architectures. A performance predictor is a function that estimates the performance of unseen architectures after being trained on a collection of architecture-performance pairs. Instead of needing to train each architecture from scratch, this allows us to predict its performance almost instantaneously (White et al., 2021, 2023). Performance predictors are based on three components: (i) the architecture-performance dataset, (ii) the prediction method, and (iii) the architecture encoding, which transforms the architecture into a suitable input format for the prediction method.

Most works combine the latter two components into a model-based prediction method (Dudziak et al., 2020). To further increase the evaluation speed during search, zero-cost proxies were introduced as target metrics for prediction methods and as the input for model-based prediction methods (Abdelfattah et al., 2021; Krishnakumar et al., 2022). More recently, FLAN (Akhauri and Abdelfattah, 2024)

combines a learned encoding of an architecture cell (Ning et al., 2023; Velickovic et al., 2018) with zero-cost-proxies, and an unsupervised learned latent space (Yan et al., 2020) as architecture encodings, which are fed into an MLP prediction. Lukasik et al. (2024) used zero-cost proxies as architecture encodings combined with a random forest prediction method for single and multi-objective tasks (accuracy and robustness). GRAF (Kadlecová et al., 2024) proposes graph features based on the architecture topology as encodings in combination with tabular predictors and shows improvements over using solely zero-cost proxies for both single and multi-objectives. These methods focus on cell-based search spaces, which are not easily transferable to other search spaces and overall lack flexibility.

In this work we are the first to show how performance prediction can speed up the search in larger, non-cell-based, more complex search spaces like einspace.

## 2.3  Large Language Models in NAS

Many studies have begun using the strong text comprehension and generation capabilities of large language models (LLMs) to tackle various aspects of NAS. LLMs can act as surrogate models to score each search candidate. In particular, GPT-4 (OpenAI, 2023) has shown promise in accelerating and potentially improving search results (Chen et al., 2024), while compact regression models distilled from GPT-4's predictions have also demonstrated comparable performance (Jawahar et al., 2024).

Moreover, LLMs—especially those designed for code generation—can be used to produce new candidate architectures by either directly outputting network structures (Yu et al., 2023; Wang et al., 2023) or serving as mutation and crossover operators (Nasir et al., 2024; Chen et al., 2023). Other research directions include pruning the search space by using GPT-4 to identify key design principles from existing architectures (Zhou et al., 2024).

Although these studies suggest that LLMs exhibit an understanding of architectural structures, most experiments have been conducted within relatively constrained or well-studied search space domains likely included in the model's training corpora. Moreover, while large, closed-source models such as GPT-4 excel as performance predictors, smaller and more cost-effective open-source models often fail to achieve comparable results (Jawahar et al., 2024). This discrepancy motivates further investigation into how open-source language models can be effectively applied to more expressive, large-scale search spaces.

## 3  Method

In this section, we present novel surrogate models for large and expressive NAS search spaces. Using surrogates to speed-up and improve search in these spaces is non-trivial due to their expressivity. We will focus our development towards the `einspace` search space as an example in this area. While most existing surrogates were designed for cell-based spaces and rely on a fixed one-hot encoding of the architecture, this is not possible in `einspace`, as the search space does not predefine the maximal depth or the number of nodes and edges in the network graph. To overcome the limits of existing surrogate-based predictors, we present two performance predictors that enable a more flexible architecture design. The first predictor is a random forest trained on a combination of zero-cost proxies and GRAF features (Kadlecová et al., 2024). In Section 3.2, we describe how we adapt this method towards `einspace` and how our implementation differs from the original cell-based GRAF variant. The second surrogate model, presented in Section 3.3, is an language model (LM)-based predictor that learns from the grammar-derived string representation of architectures.

## 3.1  Surrogate Models

Performance predictors rely on three key components: (i) an architecture-performance dataset, (ii) a prediction model, and (iii) an architecture encoding. We now formalise these components. Let $a \in \mathcal{A}$ be an architecture from the search space $\mathcal{A}$. Its true accuracy on a dataset $D$ is obtained by training and evaluating it via an expensive process, denoted as $P(a,D)$. Our goal is to approximate $P(a,D)$

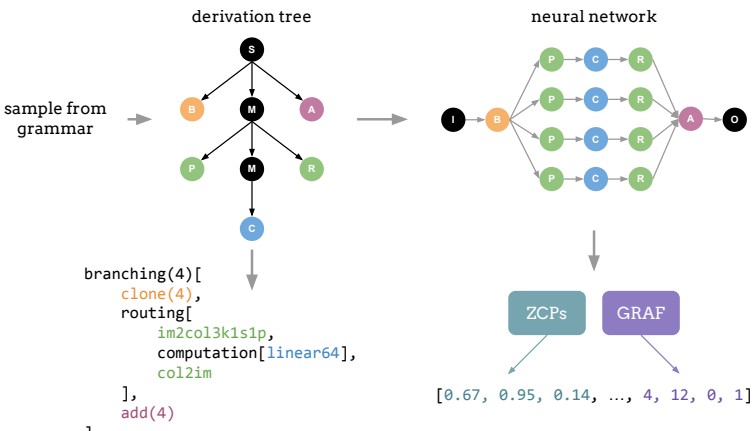

Figure 1: The encodings used by our language models (left) and our random forest and XGBoost models (right). A derivation tree is obtained from the grammar by sampling or mutation, and encoded into a string representation. The derivation tree can be compiled into a neural network, from which we can extract ZCP scores and GRAF encodings that are ultimately concatenated to form a descriptor for the RF and XGBoost models.

with a surrogate model to reduce computational cost. To achieve this, we train a model $f_\theta$ on a dataset of architecture-accuracy pairs: $S = \{(a_i, P(a_i, D))\}_{i=1}^N$. Before an architecture $a$ is input to $f_\theta$, it is first encoded using a function $E$, yielding its representation $E(a)$. The surrogate model is then trained by minimizing a loss function $\mathcal{L}$, typically the mean squared error (MSE): $\theta^* = \mathrm{argmin}_\theta \mathcal{L}(S, E; \theta)$.

## 3.2 Surrogates From GRAF and ZCP Descriptors

In the previous section, we introduced a surrogate model $f_\theta$ trained on encoded architecture-performance data. Here, we explore a specific instantiation where the encoding function $E(a)$ is constructed using a combination of topological features from GRAF and zero-cost proxies (ZCPs).

Kadlecová et al. (2024) demonstrated that tabular performance predictors—such as random forests and XGBoost—achieve state-of-the-art results on cell-based search spaces when trained on a combination of GRAF descriptors and ZCPs. However, their effectiveness in more flexible, grammar-based search spaces remains unexplored. To integrate GRAF features into our setting, we adapt them to accommodate the broader expressivity of einspace.

We include all original GRAF features, such as operation counts, max/min path lengths, and node degrees. In the original formulation, GRAF computes all possible subsets of certain features (e.g., path lengths and node degrees). However, due to the large number of operations in einspace, computing these subsets exhaustively is intractable. To address this, we redefine path lengths: the minimum path length is the shortest path containing a given operation from input to output, and the maximum path length follows a similar definition. For node degrees, we restrict our analysis to a single operation type and consider only input/output node degrees, as our initial experiments found no benefit in including average degrees. In addition to GRAF features, we incorporate the zero-cost proxies introduced by Abdelfattah et al. (2021), including grad_norm, snip, grasp, fisher, jacob_cov, plain, and synflow. The final architecture encoding, which serves as the input to our surrogate model $f_\theta$, is defined as: $E(a) = \mathrm{concat}(\mathrm{GRAF}(a), \mathrm{ZCP}(a))$

## 3.3 Language Models as Surrogates

While computing graph features and proxies on the networks can provide useful information on the performance of an architecture, we have another alternative natural representation that comes from our grammar-based search space—the *derivation tree*. This is a direct description of the architecture and its properties, that can be efficiently expressed in a string format due to its tree structure.

Therefore, we will consider using language models (LMs) directly on this string representation. To give the model more information of the inner workings of the architecture, we enrich the encoded string with additional metadata, such as the tensor sizes at the output of operations. A more detailed discussion and an ablation study of different encoding choices can be found in Section C.1. For a visual representation of this encoding and the previous, see Figure 1.

To train this surrogate, we start by initialising the parameters of the model with pre-trained language model weights. We then further fine-tune it using the following mean squared error (MSE) loss:

$$\mathcal{L}_{\text{MSE}}(\theta) = \frac{1}{|S|} \sum_{(a,P(a,D)) \in S} \left( f_\theta(E(a)) - P(a,D) \right)^2.$$ (1)

Additionally, we evaluate the few-shot learning abilities of open-source LLMs to predict the accuracy from the architecture string representation $E(a)$, with the prompts structured as *PP prompts* (Jawahar et al., 2024). Please see the appendix for more details.

### 3.4 Using Surrogates in Search

The surrogates described above will be used to speed up and improve architecture search. To do this, we build on the search method used in `einspace` (Ericsson et al., 2024), a regularised evolution algorithm (Real et al., 2019) restricted to mutation as the strategy for evolving individuals. The algorithm is modified to include our surrogates at key stages to improve the evaluation speed and selection strategy. **Improving selection**. The updated algorithm works as follows. We first randomly sample an initial population, which is evaluated and used for fitting the surrogate model. The surrogate predictions are then used in the next iteration to select the new individuals to add to the population, from a pool of $n$ mutated architectures. We use the same search routine as in `einspace` (detailed in Algorithm 1 in the appendix) with the following changes – instead of sampling 1 individual and immediately updating the population, we sample $k \geq 1$ individuals from the $n$ mutated architectures. This step is important, due to the inherent inaccuracy of the surrogates – the surrogates only approximate the accuracy, and the individual with the highest predicted accuracy is not necessarily the best one. By sampling more than 1 individual, we increase the chance of sampling the top offspring.

Along with our ZCP + GRAF + random forest and LLM surrogates, we evaluate a 'random' baseline – we create $k$ mutated offspring and add them directly to the population. This baseline is similar to the search in `einspace`, allowing for a fair comparison between a complete evaluation from scratch of each sampled individual and the introduced surrogate models.
**Replacing objective**. In addition to using surrogates for selection, we explore a surrogate-based objective function, where the surrogate fully replaces the evaluation step during search. Instead of training and evaluating architectures at every step, we rely entirely on the surrogate's predictions to rank and select individuals, thereby massively speeding up the search process. After the search has completed, the top-k running best performing architectures identified throughout the search are evaluated through full training and evaluation on the training and validation set, respectively. Then we perform the final models selection and evaluate the best model eval on the test set.

## 4 Experiments

In our experiments, we evaluate the following list of **surrogate models**.

*Random Forest:* We train a random forest (RF) regression model on the zero-cost proxy (ZCP) and GRAF feature set, as described in Section 3.2. The RF model serves as a feature-based baseline, using manually extracted architecture descriptors to predict the performance.

*BERT:* For our fine-tuning experiments, we make use of the RoBERTa (Liu et al., 2019b) and ModernBERT (Warner et al., 2024) model families. Both model families contain a base version (RoBERTa: 125M parameters, ModernBERT: 150M parameters) and a large version (RoBERTa: 355M parameters,

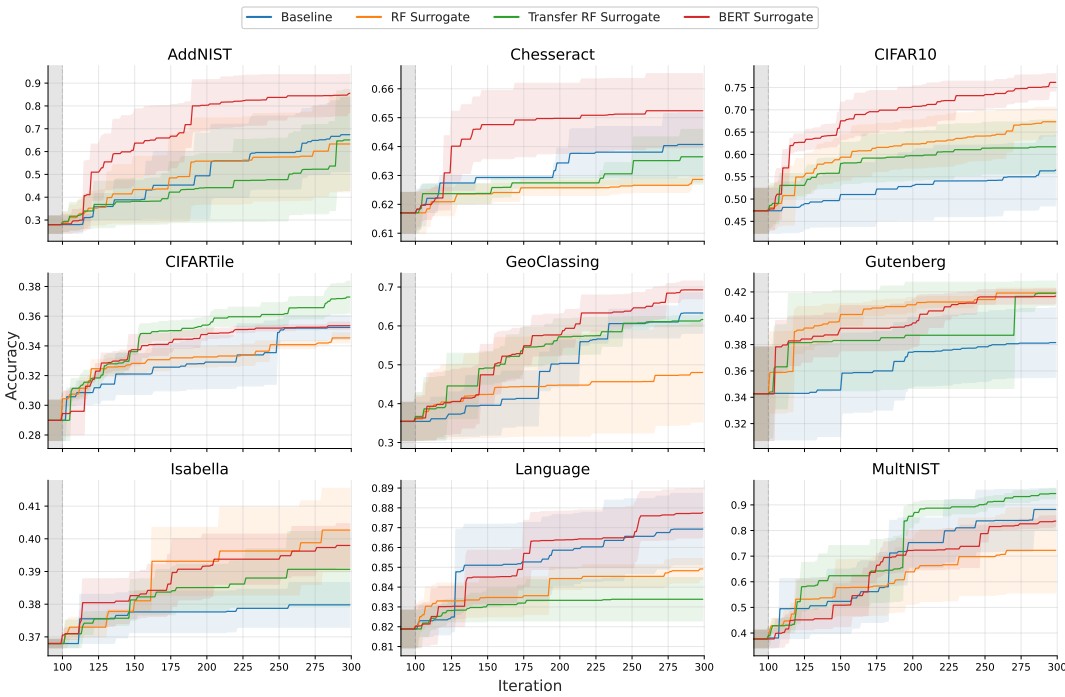

Figure 2: Search results using different surrogates. We plot the validation accuracies of the best models at each iteration of search, with the mean and standard error across three seeds.

ModernBERT: 396M parameters). We fine-tune both size variants of these models using the string representations of architectures from their derivation trees in `einspace`, as described in Section 3.3.

*Open-source LLM:* We additionally use Llama3.2 (Grattafiori et al., 2024), with its 1B, 1B Instruct, and 3B variants, as well as both Qwen2.5 (Yang et al., 2024) 7B and 14B Instruct version. We fine-tune these model in the same manner as the BERT-based models, or learn them in a few-shot manner for performance prediction.

**Evaluation settings** Assuming we have access to architecture-accuracy pairs for $N$ different datasets, we will use this data to fit and evaluate surrogate models in a few different ways.

*IID:* In this setting, we train and evaluate on data from the same dataset. This means we treat the first datapoints from a search as the training set, and the future as the evaluation set. To simulate the search setting we train on the first $m$ datapoints, predict on the next $k$. For example, we fit the first version of our surrogate on the initial population on AddNIST (Geada et al., 2024) and use it to predict the performance of the next 20 architectures. Then we can refit the surrogate every so often to evaluate throughout the search. This setting helps us to evaluate how the surrogate generalises to the immediate future of the evolution process.

*OOD/Leave-one-out:* In this setting, we train our surrogate models on $N-1$ datasets and evaluate on the single leftover dataset. As an example, we can train on all data from CIFAR10 + 7 Unseen NAS datasets and then evaluate on all data from AddNIST. This tests the surrogate's zero-shot transfer ability. In order to make this work, we need to standardise the label space (accuracy distributions) across the datasets.

*Both:* In the final setting, we combine the above two approaches. We treat all data from the other $N-1$ datasets as training data, along with any datapoints from the start of the search on the target dataset. Then as search progresses, we continue training/fine-tuning on the search data as it arrives. The implementation of this may differ between the LLM compared to RF/XGB due to the SGD vs batch training.

Table 1: Rank correlation between surrogate reward and ground truth accuracy on CIFAR10 architectures evaluation set for different surrogate models. Correlation averaged across three different seeds are given for few-shot learning surrogates with standard error. Correlation of ZCPs can be found in Table 7.

| Model | Input | Spearman | Kendall |
|---|---|---|---|
| RF | ZCP + GRAF | **0.663** | **0.514** |
| XGBRegressor | ZCP + GRAF | 0.600 | 0.449 |
| RoBERTa-base | String Encoding | 0.723 | 0.575 |
| RoBERTa-large | String Encoding | 0.679 | 0.523 |
| ModernBERT-base | String Encoding | 0.746 | 0.597 |
| ModernBERT-large | String Encoding | 0.745 | 0.612 |
| ModernBERT-large | String Encoding+aug | **0.769** | **0.628** |
| Llama3.2 1B | String Encoding | 0.722 | 0.588 |
| Llama3.2 1B Instruct | String Encoding | 0.726 | 0.593 |
| Llama3.2 3B | String Encoding | 0.682 | 0.552 |
| Llama3.2 3B Instruct | PP Prompt (3 shot) | -0.087 (0.030) | -0.067 (0.023) |
| Qwen2.5 7B Instruct | PP Prompt (3 shot) | 0.319 (0.014) | 0.235 (0.012) |
| Qwen2.5 14B Instruct | PP Prompt (3 shot) | 0.356 (0.018) | 0.256 (0.014) |

**Reported Metrics** Within each of these settings, we will report multiple metrics to understand the quality of the surrogate models. We report the rank correlation values (Spearman's rho, Kendall's tau) between the ground-truth and predicted performances. We use the surrogate model to guide the search by improving the selection of new individuals, and report the accuracy of the final architecture. For each iteration in the search, we generate a pool of $n$ candidate architectures and accept the best $k$.

In addition we also replace the objective in the actual search, and use the surrogate model directly as the objective to be optimised. Therefore, we do not include any additional training and evaluation of any networks, making this approach even faster.

**Data** In this work, we train and evaluate surrogate models using both the CIFAR10 dataset (Krizhevsky, 2009) and the tasks from the Unseen NAS benchmark (Geada et al., 2024). We use architecture-accuracy pairs obtained from searches on these datasets to form the datasets used for fitting our surrogate models. For CIFAR10, we conducted 8 NAS runs with different random seeds with same setting as in Ericsson et al. (2024) (which is also equivalent to our baseline evolution), yielding around 9k architecture-accuracy pairs. Of these 8 runs, 6 serve as training set, 1 as validation set and 1 as test set. For each of the 8 Unseen NAS tasks, we generate two runs per seed, which results in 2k pairs per dataset. One of which serves as training set and the other one as validation set.

**Search Algorithms** We run the regularised evolution variant described in Section 3.4. We use a randomly initialised initial population of 100 architectures sampled from the grammar. We set $n$ to 20 – this gives us a diverse selection of possible offspring while still being cheap, as subtree mutation can be costly for some operators in the grammar. As for the number of individuals chosen, we set $k$ to 5 as a compromise between selecting good offspring and updating the population often enough.

## 4.1 Experimental Results

**Correlation Results**. To identify the best-performing surrogate model and optimal setting for NAS searches, we evaluate rank correlations in multiple scenarios, which measures both the surrogate's ability to fit on the trained tasks, and the transferability to unseen tasks. See Appendix B for detailed implementation.

*Correlations on CIFAR10:* We first focus on the case in which we train and evaluate the surrogate on CIFAR10 data, to assess the model's ability to capture and generalise to new architectures within the same dataset. For the few-shot learning settings, examples are chosen uniformly based on the accuracy from training set. As shown in Table 1, the fine-tuned ModernBERT-large achieves the highest correlation on the evaluation set, which can be further improved by adding data augmentation (cf. Appendix B). Among the Llama3.2 models in the fine-tuning setting, the 1B and

Table 2: Kendall-Tau correlation on the Unseen NAS datasets for surrogate ModernBERT-large and random forest. For CIFAR10 we use 7k architectures to train the surrogate on, while on the other datasets we use 1 000. For a fair comparison, one seed is randomly chosen from CIFAR10 training set to be comparable to other tasks.

| | CIFAR10 | AddNIST | Language | MultNIST | CIFARTile | Gutenberg | Isabella | GeoClassing | Chesseract | Avg |
|---|---|---|---|---|---|---|---|---|---|---|
| CIFAR10 | **0.612** / **0.648** | 0.470 / 0.516 | **0.408** / 0.326 | **0.604** / 0.447 | **0.470** / 0.374 | 0.577 / 0.525 | 0.189 / 0.239 | **0.581** / 0.284 | 0.280 / 0.424 | **0.466** / **0.420** |
| CIFAR10(1k) | 0.524 / 0.582 | 0.479 / 0.370 | 0.374 / 0.380 | 0.507 / 0.434 | 0.320 / **0.379** | 0.439 / **0.586** | 0.254 / 0.236 | 0.360 / 0.375 | 0.157 / 0.414 | 0.379 / 0.417 |
| AddNIST | 0.331 / 0.526 | **0.589** / **0.577** | 0.271 / 0.342 | 0.460 / **0.478** | 0.387 / 0.336 | 0.362 / 0.577 | 0.223 / 0.163 | 0.459 / 0.230 | 0.179 / 0.412 | 0.362 / 0.405 |
| Language | 0.370 / 0.486 | 0.443 / 0.229 | 0.388 / 0.200 | 0.463 / 0.128 | 0.393 / 0.126 | 0.467 / 0.133 | 0.268 / 0.268 | 0.262 / 0.219 | 0.271 / **0.430** | 0.369 / 0.246 |
| MultNIST | 0.371 / 0.497 | 0.369 / 0.280 | 0.264 / **0.401** | 0.394 / 0.445 | 0.285 / 0.307 | 0.423 / 0.500 | 0.229 / 0.241 | 0.245 / 0.307 | 0.300 / 0.335 | 0.320 / 0.368 |
| CIFARTile | 0.233 / 0.553 | 0.522 / 0.575 | 0.114 / 0.236 | 0.336 / 0.365 | 0.397 / 0.332 | 0.286 / 0.495 | 0.212 / -0.056 | 0.484 / 0.219 | 0.256 / 0.257 | 0.316 / 0.331 |
| Gutenberg | 0.232 / -0.055 | 0.165 / 0.425 | 0.404 / 0.358 | 0.345 / 0.315 | 0.163 / 0.234 | 0.411 / 0.295 | 0.191 / 0.225 | 0.118 / 0.205 | 0.306 / 0.360 | 0.259 / 0.262 |
| Isabella | 0.250 / 0.054 | 0.364 / 0.174 | 0.305 / 0.184 | 0.440 / 0.160 | 0.281 / 0.233 | 0.278 / 0.414 | 0.240 / **0.298** | 0.244 / **0.381** | 0.202 / 0.218 | 0.267 / 0.235 |
| GeoClassing | 0.277 / 0.559 | 0.506 / 0.547 | 0.296 / 0.251 | 0.361 / 0.464 | 0.407 / 0.315 | 0.424 / 0.578 | **0.285** / 0.233 | 0.437 / 0.007 | 0.136 / 0.316 | 0.313 / 0.363 |
| Chesseract | 0.285 / 0.254 | 0.142 / -0.020 | 0.220 / 0.253 | 0.363 / 0.216 | 0.246 / 0.226 | 0.432 / 0.128 | 0.116 / 0.158 | -0.021 / 0.089 | **0.404** / 0.405 | 0.243 / 0.190 |

1B Instruct models also achieve moderately high correlations. However, few-shot learning models result in noticeable drop in performance. The ZCPs perform poorly overall, with `jacov_cov` and `synflow` being the strongest ones. They still lag behind the more complex models, showing the need for more complex heuristics in this expressive search space. Tree-based models trained on ZCPs and GRAF give moderate correlations which are lower compared to the best-performing LMs; however, they require no pre-training, which is an advantage. Based on these results, we choose ModernBERT-large and the random forest regression as the surrogate models going forward.

*Transfer Across Tasks:* As shown in Table 2, training on the 1 000 samples from CIFAR10 gives overall the highest transfer correlations among that group. This may be due to its generic image classification task or a greater diversity compared to other datasets. Due to this we also consider training models with more data from CIFAR10, for a total of 7k samples. From this data, we see ModernBERT-large reaching the highest average correlation (0.466). CIFAR10-trained surrogates excel on MultNIST (0.604), Gutenberg (0.577), and GeoClassing (0.581), demonstrating strong cross-task generalisation. Language-based datasets like Gutenberg transfer well within their domain (0.404 on Language) but struggle with vision tasks, highlighting modality transfer challenges. Isabella and Chesseract yield the lowest correlations, indicating limited generalisation. ModernBERT-large consistently outperforms the random forest model, underscoring the advantage of deep LMs in capturing complex relationships. However, tree-based models remain competitive in well-aligned tasks, such as CIFARTile to CIFAR10 (0.553).

**Leave-one-out Correlations:** We now assess how well surrogates trained on all but one dataset generalise to the excluded task (Table 3). ModernBERT-large achieves the highest Kendall correlation on Gutenberg (0.658) and MultNIST (0.625), reinforcing its ability to generalise across diverse tasks. With more data available for training, we observe a general improvement in Kendall correlations across tasks. Compared to the single-dataset transfer setting, leave-one-out training leads to higher correlations, particularly on complex tasks like Gutenberg (0.658 vs. 0.577) and MultNIST (0.625

Table 3: Correlation on the Unseen NAS datasets for Leave-one-out surrogates with percentile normalisation.

| Eval Tasks | ModernBERT-large | | Random Forest | |
|---|---|---|---|---|
| | Spearman | Kendall | Spearman | Kendall |
| AddNIST | 0.773 | 0.625 | 0.671 | 0.500 |
| Language | 0.610 | 0.433 | 0.623 | 0.450 |
| MultNIST | 0.798 | 0.625 | 0.722 | 0.533 |
| CIFARTile | 0.629 | 0.444 | 0.558 | 0.374 |
| Gutenberg | 0.835 | 0.658 | 0.816 | 0.643 |
| Isabella | 0.273 | 0.186 | 0.308 | 0.211 |
| GeoClassing | 0.661 | 0.475 | 0.693 | 0.504 |
| Chesseract | 0.545 | 0.384 | 0.599 | 0.423 |

vs. 0.604) for ModernBERT-large. However, Isabella remains a challenging dataset (0.186–0.211), indicating that increasing training diversity does not always guarantee better generalization if the task is inherently misaligned.

**Search Results**. We now present our results from using the surrogate models to guide search. From Table 4 we see that surrogate-assisted evolution generally outperforms the baseline across most tasks, with ModernBERT-large (Evolution(BERT)) achieving the highest average accuracies. This suggests that deep learning-based surrogates provide more effective guidance during architecture search compared to both random selection and tree-based models like the Random Forest (RF). The Evolution(BERT) variant outperforms the baseline evolution approach on nearly every task, with particularly strong gains on CIFAR10, AddNIST, Language, and Isabella, demonstrating the model's ability to generalize across diverse datasets.

The Random Forest-based approaches (Evolution(RF) and Evolution(RF Transfer)) show mixed results, with Evolution(RF) performing better than the baseline in some cases (e.g., CIFAR10 and Gutenberg) but struggling in others. The transfer learning variant (RF Transfer) does well on MultNIST but is less consistent overall, suggesting that while transfer learning can help in some scenarios, it does not always lead to better search performance.

In Figure 2, we show the validation accuracies of the best models at each point in the search runs. We can see from this that our surrogate-based searches often continuously outperforms the baseline> Furthermore, the best baseline performance is often reached in many fewer iterations, highlighting that the surrogates greatly improve search efficiency. This is most prominent on CIFAR10, Gutenberg and Isabella, where all our surrogates improve upon the baseline. The ModernBERT-large variant also shines on AddNIST and Chesseract where it dominates the others significantly.

The final row of Table 4 show the performance when using the ModernBERT-large variant directly as the search objective. This method is competitive in some instances, even outperforming the baseline Evolution on CIFAR10 and Chesseract. However, it is still far behind the best surrogate-guided search. We are interested to see how the performance of this style of method improves with better future surrogate models, and its potential use for initialising search populations.

Overall, these results show that surrogate-assisted evolution significantly enhances architecture search performance, with ModernBERT-large proving to be the strongest surrogate model. It consistently outperforms the baseline evolution approach, demonstrating the value of deep learning models for guiding search. Random Forest-based surrogates provide some benefits, but their effectiveness varies depending on the dataset. While using the surrogate directly as the search objective does not yet match full evaluations, its competitive performance on some tasks suggests promise for future improvements in surrogate-driven search methods.

Table 4: Search results using regularised evolution on the Unseen NAS datasets. For each version we run on 3 random seeds and report the average best accuracy found along with the standard error of the mean. Refer to Appendix B.4 for detailed evolution settings.

| | CIFAR10 | AddNIST | Langauge | MultNIST | CIFARTile | Gutenberg | Isabella | GeoClassing | Chesseract |
|---|---|---|---|---|---|---|---|---|---|
| Evolution | 0.624(0.096) | 0.706(0.160) | 0.899(0.020) | 0.841(0.080) | **0.358**(0.002) | 0.406(0.039) | 0.449(0.038) | 0.725(0.030) | 0.595(0.016) |
| Evolution(RF) | 0.735(0.025) | 0.591(0.197) | 0.881(0.006) | 0.690(0.160) | 0.331(0.004) | **0.433**(0.009) | 0.470(0.013) | 0.497(0.117) | 0.601(0.014) |
| Evolution(RF Transfer) | 0.680(0.063) | 0.656(0.229) | 0.860(0.015) | **0.877**(0.070) | 0.336(0.012) | 0.422(0.006) | 0.451(0.029) | 0.687(0.023) | 0.595(0.012) |
| Evolution(BERT) | **0.828**(0.006) | **0.846**(0.088) | **0.910(0.017)** | 0.765(0.019) | 0.341(0.010) | 0.425(0.005) | **0.512**(0.040) | **0.735**(0.024) | **0.606**(0.008) |
| Evolution(BERT as obj) | 0.656(0.046) | 0.479(0.165) | 0.857(0.013) | 0.557(0.169) | 0.290(0.007) | 0.359(0.038) | 0.440(0.029) | 0.512(0.061) | 0.597(0.028) |

# 5 Conclusion

In this paper, we have demonstrated the effectiveness of surrogate models, particularly fine-tuned large language models (LLMs), in accelerating neural architecture search (NAS) within expressive search spaces. Our results show that these surrogates can significantly reduce search costs, with performance predictors helping to guide evolutionary search and improve efficiency across diverse tasks. However, the overall performance of the searches in `einspace` can still be improved, especially

from randomly seeded population. We think there is room for pushing this further by leveraging these surrogate models. Additionally, continued development of performance predictors, combined with the advancements in LLMs, promises even stronger improvements in search efficiency. As these techniques evolve, we anticipate a future where NAS can more effectively explore large, complex search spaces, uncovering novel and high-performing architectures with greater speed and precision.

**Limitations**. Our inclusion of surrogates in large, expressive search spaces reduced search costs by minimizing the need for full evaluations. We adapted a high-performing prediction method (GRAF) and incorporated large language models, but there is still room for optimization through custom evolutionary operators. This paper focuses on a single objective (image classification accuracy), but future work will explore multi-objective searches (e.g., hardware, robustness) and hardware-aware surrogates. While our prediction method shows promising speed-ups, it does not outperform baseline search on all unseen data, a challenge for future research.

**Broader Impact**. Search in expressive search spaces is inherently more expensive than in limited search spaces, which is a potential negative effect on the environment. However, discovering novel architectures with an efficient design could have a positive effect on both humanity and the environment. To get there, it is essential to reduce the high training and evaluation costs. We believe our method brings us closer to this goal – future work can focus on improving the surrogates we introduced, and lessen the evaluation costs even more.

**Acknowledgements**. This work was supported by the Engineering and Physical Sciences Research Council [EP/X020703/1], a studentship from the School of Engineering at the University of Edinburgh and SVV project number 260 821. Computational resources were provided by the e-INFRA CZ project (ID:90254), supported by the Ministry of Education, Youth and Sports of the Czech Republic. JL acknowledges support by the German Research Foundation research unit 5336 Learning to Sense and by the Lamarr Institute for Machine Learning and Artificial Intelligence.

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

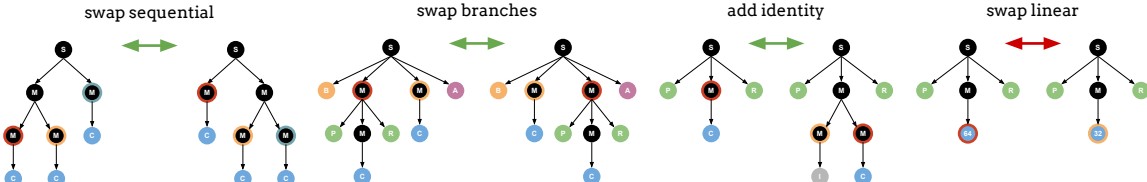

Figure 3: We apply four forms of augmentation to the architecture dataset to increase the effective data size for LM training. Three of our augmentations preserve the same exact network, while the last causes minor changes: (1) We swap the nesting order of `sequential` models; (2) We swap the two branches in a `branching(2)` module; (3) We add components that reduce to the `identity` operation; and (4) We change the output dimensionality to a neighbouring option in the list `[16, 32, 64, 128, 256, 512, 1024, 2048]`. For each augmented architecture, we adjust the corresponding accuracy by adding Gaussian noise from $\mathcal{N}(0, 0.005^2)$.

## A  Algorithmic Details

### A.1  Architecture Augmentation

To enhance generalisation and increase the effective dataset size for LM training, we apply four forms of data augmentation to our architecture dataset. These augmentations aim to provide diverse yet functionally equivalent architectures:

1. Reordering Sequential Modules – We swap the nesting order of operations within `sequential` models while maintaining the same computation.

2. Reordering Branching Modules – For architectures containing `branching(2)` modules, we swap the order of the branches without affecting functionality.

3. Identity-Preserving Modifications – We introduce additional components that mathematically reduce to the `identity` operation, ensuring no impact on functionality while diversifying representations.

4. Perturbing Output Dimensionality – We slightly modify the output tensor size by selecting a neighbouring value from the list `[16, 32, 64, 128, 256, 512, 1024, 2048]`. This augmentation changes the the architecture functionally, although from our experiments the change is relatively minor.

To maintain some diversity in the label space we also adjust the corresponding accuracy of augmented architectures by a very small amount, by adding Gaussian noise sampled from $\mathcal{N}(0, 0.005^2)$. These augmentations provide synthetic diversity, helping the LM surrogate learn a more robust mapping from architecture representations to performance estimates. Figure 3 visualises these augmentation techniques.

## B  Implementation Details

### B.1  Random Seeds

We use random seeds 0 through 7 to generate architecture-accuracy pairs (for tasks that only require two seeds, we use seeds 0 and 1). For the actual searches, we use seeds 42, 43, and 44.

### B.2  Language Models Implementation Details

Settings for fine-tuning LMs include: learning rate set to $10^{-5}$, batch size to 2, weight decay to 0.01, cosine learning rate scheduler and 0.06 warmup ratio. We train the model for 5 epochs on

**Algorithm 1** Regularised Evolution with Mutation and a Surrogate

---

**Input**: Architecture space $\mathcal{A}$, sampling function SAMPLE : $\{*\} \to \mathcal{A}$, mutation function MUTATE : $\mathcal{A} \to \mathcal{A}$, evaluation function TRAINANDEVAL : $\mathcal{A} \to \mathbb{R}^+$, surrogate function SURROGATE : $\mathcal{A} \to \mathbb{R}^+$, population size $p$, tournament size $\tau$, number of offspring samples $c$, number of chosen offspring per iteration $k$, and number of iterations $n$.

**Output**: `best_individual.arch`

```
 1: population = ∅
 2: for i = 1 to p do
 3:     individual.arch = SAMPLE()                          ▷ Sample random architecture
 4:     individual.accuracy = TRAINANDEVAL(individual.arch)
 5:     add individual to population
 6: end for
 7: for i = p+1 to n do
 8:     offspring = ∅
 9:     for j = 1 to c do
10:         parent = TOURNAMENTSELECTION(population, τ)
11:         child.arch = MUTATE(parent.arch)
12:         child.prediction = SURROGATE(child.arch)           ▷ Predict performance (cheap)
13:         add child to offspring
14:     end for
15:     offspring = TOPK(offspring, k)             ▷ Select best based on predicted performance
16:     for child in offspring do
17:         child.accuracy = TRAINANDEVAL(child.arch)       ▷ Actual performance (expensive)
18:         add child to population
19:         pop oldest individual from population                              ▷ Aging
20:     end for
21:     Fit SURROGATE function on population          ▷ Continuously train the surrogate
22: end for
23: best_individual = argmax_{individual∈population} individual.accuracy
```

---

**Algorithm 2** TOURNAMENTSELECTION

---

**Input**: Population of architectures `population`, tournament size $\tau$

**Output**: Selected `individual`

```
 1: tournament ← RANDOMSUBSET(population, τ)                        ▷ Uniformly at random
 2: individual = argmax_{individual∈tournament} individual.accuracy           ▷ Return best
```

---

leave-one-out experiments and 15 epochs on all other experiments. The best checkpoint is saved based on Kendall Tau correlation on the evaluation set.

For few-shot learning, we set the temperature to 0.7 and maximum new tokens to 50, the first floating number is extracted as prediction; if no floating number is found, we retry with maximum new tokens as 500. The prompt follows the structure of the PP prompts (Jawahar et al., 2024). Prompt details are presented in Appendix E.

### B.3 Random Forest (XGBoost) Implementation Details

For the random forest regressor, we used the default settings of scikit-learn (Pedregosa et al., 2011). For the XGBoost Regressor model, we used optimised parameters from Autogluon (Erickson et al., 2020).

### B.4 Evolutionary Algorithm Hyperparameters

The evolutionary algorithm generates 20 independently mutated individuals per iteration. The surrogate variants use a model to rank and chooses the top 5 to fully train and test and include in the population, while the base version chooses 5 randomly. The population size is 100, and the search runs for 300 iterations [1]. For each version, we run on 3 random seeds.

The BERT based surrogate model is re-fit every 100 iterations, the random forest based models are refit every 20 iterations.

### B.5 Compute Resources

All our experiments ran on 7 clusters with the following infrastructure:

- AMD EPYC 7552 48-Core Processor with 1000GB RAM and 8 × NVIDIA RTX A5500 with 24GB of memory

- AMD EPYC 7262 8-Core Processor with 125GB RAM and 7 × NVIDIA A100 with 40GB of memory

- AMD EPYC 7543 64-Core Processor with 512GB RAM and 4 × NVIDIA A40 with 48GB of memory

- 2 × AMD EPYC 9454 48-Core Processor with 1536GB RAM and 2 × NVIDIA H100 with 94GB of memory

- 2 × AMD EPYC 7662 64-Core Processor with 1000GB RAM and 4 × NVIDIA A100 with 40GB of memory

- 2 × AMD EPYC 9554 64-Core Processor with 1536GB RAM and 2 × NVIDIA L40 with 48GB of memory

- 2 × AMD EPYC 7513 32-Core Processor with 512GB RAM and 1 × NVIDIA A40 with 48GB of memory

## C  Ablation Study

### C.1  Encoding for LMs

We conducted an ablation study on three different encodings used as input to the language model:

---

[1] Shortly before the deadline, we found and fixed an inconsistency (different scaling of synflow values) in the handling of transfer learning data compared to the data from the current run in the transfer version of the RF surrogate. We were unable to re-run 7 out of the 27 experiments to the full 300 iterations (two seeds on addnist and multnist, one seed on chesseract, gutenberg, geoclassing, and language). All of the runs achieved at least 200 iterations. For these runs, we use the best value they found as the value for the rest of the iterations. We will update the results for the final version of the paper.

Table 5: Correlation on CIFAR10 architectures evaluation set for different architecture encodings.

| Encoding | Spearman | Kendall |
|---|---|---|
| Derivation tree str | 0.621 | 0.448 |
| Derivation tree str (+ shape) | **0.745** | **0.612** |
| PyTorch str | 0.698 | 0.535 |

- **Derivation tree string**
  The string representation of architecture's derivation tree (e.g., `routing[im2col(3,2,1)`, `computation<linear(128)>, col2im]`

- **Derivation tree string + shape**
  Derivation tree string augmented with output feature shape information (e.g., `routing[im2col(3,2,1) {'out_feature_shape': [256, 3]} computation<linear(128)> {'out_feature_shape': [256, 32]}, col2im {'out_feature_shape': [32, 16, 16]}])`

- **PyTorch modules**
  The string representation of built PyTorch model. (e.g., `nn.Conv2d(out_channels=128, kernel_size=3, stride=2, padding=1))`

We evaluated these encodings on the CIFAR10 architecture dataset using ModernBERT-large as the surrogate, and the results are summarised in Table 5. The derivation tree string **with shape information** achieved the highest Spearman (0.745) and Kendall (0.612) correlation, outperforming both the simple derivation tree string and the PyTorch modules representation. These findings suggest that including feature shape information is crucial for capturing architecture characteristics that help the language model make more accurate predictions. Furthermore, the strong influence of architecture encodings on surrogate performance highlights the potential for further improvements through better encoding strategies.

## C.2 Normalization Methods

The model accuracies have different ranges on different datasets this may affect the performance of the surrogate models, therefore, we considered different ways how to aggregate the data:

**minmax**. The target values from all the datasets are first scaled between 0 and 1. Only then the datasets are merged to create a single training set.

**percentile**. The target values from all the datasets are first replaced by their percentiles among the values on the same dataset (essentially normalizing their ranks to 0-1). Only then the datasets are merged.
The ablation of normalization methods for surrogate models are shown in Table 6.

## D  Additional Results

**Search with longer iteration**
To see whether the surrogate keeps improving search for longer runs, we continue our searches to 500 iterations for the AddNIST and Language datasets, comparing our EvolutiON(BERT) baseline models. Figure 4 shows how in the case of AddNIST, we see continuous improvements until the validation accuracies cap out around 100%, while on Language we see moderate continued improvement. Overall this suggests the surrogates can help throughout long searches, though we hypothesise that they will provide the strongest boosts in the beginning and middle of search.

Table 6: Leave-one-out surrogate correlations for different normalization methods.

| | | BERT | | Random Forest | | XGB Regressor | |
| | | Spearman | Kendall | Spearman | Kendall | Spearman | Kendall |
| Dataset | Normalisation | | | | | | |
|---|---|---|---|---|---|---|---|
| AddNIST | minmax | 0.768 | 0.617 | 0.599 | 0.433 | 0.541 | 0.397 |
| CIFARTile | minmax | 0.609 | 0.433 | 0.494 | 0.327 | 0.563 | 0.393 |
| AddNIST | percentile | 0.773 | 0.624 | 0.671 | 0.500 | 0.551 | 0.409 |
| CIFARTile | percentile | 0.629 | 0.444 | 0.558 | 0.374 | 0.557 | 0.388 |

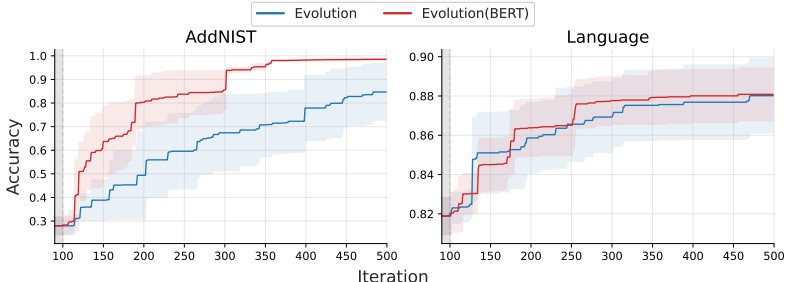

Figure 4: Extended runs for ModernBERT-large surrogates.

Table 7: Rank correlation between ZCP reward and ground truth accuracy on CIFAR10 architectures evaluation set.

| ZCP | Spearman | Kendall |
|---|---|---|
| grad_norm | 0.040 | 0.036 |
| snip | 0.094 | 0.078 |
| grasp | -0.072 | -0.053 |
| fisher | -0.037 | -0.020 |
| jacob_cov | 0.381 | 0.243 |
| plain | -0.189 | -0.126 |
| synflow | 0.465 | 0.362 |

**Continual training during search**: We evaluate how the correlations change when we continuously refit or fine-tune the models as data comes in during the search. Table 8 shows that continued training helps the BERT, RF and XGBoost models significantly. We refit every 100 iterations.

Table 8: Continual training results with "✓" indicates the model was trained during evaluation; "✗" means it was not. We use BERT in short for ModernBERT-large and report average the correlation across each 100 iterations across 2 different seeds.

| Model | Train | AddNIST | Langauge | MultNIST | CIFARTile | Gutenberg | Isabella | GeoClassing | Chesseract | Avg |
|---|---|---|---|---|---|---|---|---|---|---|
| BERT | ✓ | 0.203 | 0.412 | 0.288 | 0.273 | 0.416 | 0.233 | 0.383 | 0.390 | 0.325 |
| BERT + Cifar10 | ✓ | 0.386 | 0.557 | 0.510 | 0.380 | 0.553 | 0.423 | 0.581 | 0.541 | 0.491 |
| | ✗ | 0.258 | 0.313 | 0.396 | 0.274 | 0.350 | 0.052 | 0.381 | 0.199 | 0.278 |
| BERT + 8 Others | ✓ | 0.394 | 0.467 | 0.513 | 0.398 | 0.558 | 0.391 | 0.582 | 0.494 | 0.475 |
| | ✗ | 0.318 | 0.296 | 0.418 | 0.264 | 0.431 | 0.111 | 0.259 | 0.284 | 0.298 |
| RF | ✓ | 0.246 | 0.551 | 0.393 | 0.360 | 0.550 | 0.472 | 0.521 | 0.561 | 0.457 |
| RF + Cifar10 | ✓ | 0.250 | 0.494 | 0.367 | 0.383 | 0.480 | 0.461 | 0.489 | 0.500 | 0.428 |
| | ✗ | 0.234 | 0.272 | 0.280 | 0.271 | 0.327 | 0.032 | 0.291 | 0.332 | 0.255 |
| RF + 8 Others | ✓ | 0.263 | 0.437 | 0.362 | 0.368 | 0.471 | 0.480 | 0.492 | 0.526 | 0.425 |
| | ✗ | 0.246 | 0.286 | 0.304 | 0.297 | 0.397 | 0.187 | 0.312 | 0.373 | 0.300 |
| XGB | ✓ | 0.253 | 0.560 | 0.404 | 0.350 | 0.543 | 0.474 | 0.530 | 0.563 | 0.460 |
| XGB + Cifar10 | ✓ | 0.279 | 0.507 | 0.376 | 0.380 | 0.485 | 0.470 | 0.517 | 0.519 | 0.442 |
| | ✗ | 0.244 | 0.266 | 0.213 | 0.306 | 0.341 | 0.029 | 0.246 | 0.322 | 0.246 |
| XGB + 8 Others | ✓ | 0.298 | 0.516 | 0.378 | 0.376 | 0.510 | 0.480 | 0.499 | 0.533 | 0.449 |
| | ✗ | 0.260 | 0.336 | 0.315 | 0.246 | 0.429 | 0.119 | 0.290 | 0.379 | 0.297 |

Table 9: Correlation on the Unseen NAS datasets for Leave-one-out surrogates with percentile normalisation.

| Eval Tasks | ModernBERT-large | | Random Forest | | XGB Regressor | |
|---|---|---|---|---|---|---|
| | Spearman | Kendall | Spearman | Kendall | Spearman | Kendall |
| AddNIST | 0.773 | 0.625 | 0.671 | 0.500 | 0.551 | 0.409 |
| Language | 0.610 | 0.433 | 0.623 | 0.450 | 0.622 | 0.456 |
| MultNIST | 0.798 | 0.625 | 0.722 | 0.533 | 0.749 | 0.561 |
| CIFARTile | 0.629 | 0.444 | 0.558 | 0.374 | 0.557 | 0.388 |
| Gutenberg | 0.835 | 0.658 | 0.816 | 0.643 | 0.839 | 0.669 |
| Isabella | 0.273 | 0.186 | 0.308 | 0.211 | 0.278 | 0.187 |
| GeoClassing | 0.661 | 0.475 | 0.693 | 0.504 | 0.667 | 0.475 |
| Chesseract | 0.545 | 0.384 | 0.599 | 0.423 | 0.626 | 0.445 |

## D.1 Transfer Learning Surrogate Selection

We consider two different possible machine learning models for the transfer surrogates —- the XGBoost regressor and the random forest regressor. In order to decide which of them to use, we evaluated them in different scenarios using the static data obtained from a number of evolutionary runs. We were interested to see how they perform under different conditions, both in cases where we use them separately only on the single dataset (in the RF surrogate runs), and together with data from the other datasets in the transfer learning settings. In this section, for the transfer learning experiments, we consider cases where only the cifar10 data are used, only the data from the other Unseen datasets are used, or all the data are used.

Additionally, we evaluated the difference in the cases when the model was trained only once using the transfer data and used for the whole run, or when it was retrained after every 100 iterations to predict the next 100 values.

Finally, in some rare cases some networks cannot be evaluated, these are assigned zero accuracy. We investigate, how keeping them or removing them affects the final performance of the models.

The results of these experiments are summarized in Tables 13-16. For brevity and consistency with the language model tables, we show only the combinations of parameters where training on

Table 10: Kendall-Tau correlation on the Unseen NAS datasets for the XGBoost regressor surrogate

| | CIFAR10 | AddNIST | Language | MultNIST | CIFARTile | Gutenberg | Isabella | GeoClassing | Chesseract | Avg |
|---|---|---|---|---|---|---|---|---|---|---|
| CIFAR10 | 0.639 | 0.466 | 0.349 | 0.459 | 0.381 | 0.599 | 0.281 | 0.174 | 0.405 | 0.417 |
| CIFAR10(1k) | 0.535 | 0.187 | 0.365 | 0.409 | 0.391 | 0.506 | 0.254 | 0.354 | 0.332 | 0.370 |
| AddNIST | 0.488 | 0.596 | 0.337 | 0.489 | 0.423 | 0.503 | 0.248 | 0.261 | 0.421 | 0.418 |
| Language | 0.443 | 0.125 | 0.277 | 0.216 | 0.087 | 0.205 | 0.255 | 0.306 | 0.386 | 0.255 |
| MultNIST | 0.495 | 0.330 | 0.366 | 0.416 | 0.301 | 0.450 | 0.183 | 0.153 | 0.304 | 0.333 |
| CIFARTile | 0.523 | 0.508 | 0.247 | 0.368 | 0.280 | 0.437 | -0.082 | 0.105 | 0.222 | 0.290 |
| Gutenberg | -0.091 | 0.262 | 0.298 | 0.293 | 0.098 | 0.277 | 0.153 | 0.115 | 0.309 | 0.190 |
| Isabella | -0.006 | 0.150 | 0.224 | 0.332 | 0.362 | 0.364 | 0.225 | 0.390 | 0.233 | 0.253 |
| GeoClassing | 0.544 | 0.549 | 0.177 | 0.466 | 0.302 | 0.557 | 0.245 | 0.118 | 0.340 | 0.367 |
| Chesseract | 0.046 | 0.019 | 0.239 | 0.287 | 0.237 | 0.172 | 0.143 | 0.074 | 0.372 | 0.177 |

cifar is performed and additionally with removing of zero accuracy networks. All the tables show Kendall tau between the predicted values for the next 100 iterations and the real values.

The performance of both the models aggregated over all the different number of iterations for all datasets is in Table 8. Aggregation over different datasets together with different normalization techniques and removing or not removing zero accuracy networks for all the different iterations is in Tables 17 and 18.

In these tables, we can see that removing the zero accuracy networks has almost no effect in the case without transfer learning (Table 17). In the transfer learning case (cf. Table 18) the effect of removing these networks is almost always positive. The different normalization methods again do not have any significant effect in the case without transfer learning, but significantly improve the results in the transfer learning case. The differences between the two methods are rather small when compared over the different number of iterations, but in the initial part (after 100 iterations) the minmax seems to be slightly better than the percentile normalization.

The difference between the two models is also quite small when evaluated over the different number of iterations, but random forest seems to be slightly better in the initial phase (after 100 iterations). Later, XGBoost tends to be a bit better.

Based on these comparisons, and also the fact that our optimization runs are relatively short with only 300 iterations, we decided to use random forest without removing the zero-accuracy networks and any normalization for the baseline run without transfer learning, and random forest with removing zero-cost networks and minmax normalization for the transfer learning experiments. For longer runs, using XGBoost might be beneficial, although we believe that the performance in the beginning of the search may be still more important than the relatively small difference in the performance later.

Table 11: Results of the ModernBERT-large models in simulated surrogate runs, seed=0

| Task | Model | Train | 1 | 2 | 3 | 4 | 5 | 6 | 7 | 8 | 9 | Avg |
|------|-------|-------|---|---|---|---|---|---|---|---|---|-----|
| AddNIST | BERT | ✓ | 0.11 | 0.41 | 0.16 | 0.02 | 0.46 | 0.13 | 0.06 | 0.21 | 0.09 | 0.18 |
| | BERT + Cifar10 | ✓ | 0.66 | 0.62 | 0.42 | 0.37 | 0.66 | 0.36 | 0.27 | 0.38 | 0.28 | 0.45 |
| | | ✗ | 0.54 | 0.60 | 0.31 | 0.27 | 0.44 | 0.12 | -0.01 | 0.28 | 0.14 | 0.30 |
| | BERT + 8 Others | ✓ | 0.66 | 0.62 | 0.39 | 0.34 | 0.58 | 0.40 | 0.27 | 0.40 | 0.28 | 0.44 |
| | | ✗ | 0.71 | 0.62 | 0.36 | 0.15 | 0.56 | 0.18 | 0.14 | 0.34 | 0.24 | 0.37 |
| Chesseract | BERT | ✓ | 0.41 | 0.33 | 0.26 | 0.38 | 0.45 | 0.65 | 0.38 | 0.40 | 0.51 | 0.42 |
| | BERT + Cifar10 | ✓ | 0.33 | 0.50 | 0.54 | 0.62 | 0.55 | 0.76 | 0.57 | 0.57 | 0.63 | 0.56 |
| | | ✗ | 0.13 | 0.07 | 0.17 | 0.23 | 0.10 | 0.09 | 0.07 | 0.08 | 0.16 | 0.12 |
| | BERT + 8 Others | ✓ | 0.24 | 0.50 | 0.45 | 0.41 | 0.47 | 0.67 | 0.55 | 0.52 | 0.59 | 0.49 |
| | | ✗ | 0.26 | 0.30 | 0.31 | 0.34 | 0.33 | 0.34 | 0.21 | 0.16 | 0.17 | 0.27 |
| CIFARTile | BERT | ✓ | 0.39 | 0.18 | 0.19 | 0.23 | 0.41 | 0.29 | 0.22 | 0.33 | 0.34 | 0.29 |
| | BERT + Cifar10 | ✓ | 0.43 | 0.41 | 0.46 | 0.37 | 0.59 | 0.26 | 0.48 | 0.49 | 0.45 | 0.44 |
| | | ✗ | 0.41 | 0.33 | 0.27 | 0.30 | 0.32 | 0.30 | 0.41 | 0.34 | 0.32 | 0.33 |
| | BERT + 8 Others | ✓ | 0.44 | 0.38 | 0.56 | 0.49 | 0.58 | 0.30 | 0.41 | 0.41 | 0.53 | 0.46 |
| | | ✗ | 0.41 | 0.41 | 0.25 | 0.25 | 0.26 | 0.21 | 0.39 | 0.35 | 0.33 | 0.32 |
| GeoClassing | BERT | ✓ | 0.18 | 0.35 | 0.14 | 0.15 | 0.31 | 0.28 | 0.42 | 0.48 | 0.46 | 0.31 |
| | BERT + Cifar10 | ✓ | 0.32 | 0.46 | 0.59 | 0.65 | 0.71 | 0.52 | 0.75 | 0.76 | 0.68 | 0.60 |
| | | ✗ | 0.33 | 0.41 | 0.47 | 0.37 | 0.35 | 0.31 | 0.47 | 0.60 | 0.49 | 0.42 |
| | BERT + 8 Others | ✓ | 0.37 | 0.19 | 0.53 | 0.63 | 0.74 | 0.56 | 0.78 | 0.78 | 0.68 | 0.58 |
| | | ✗ | 0.12 | 0.12 | 0.33 | 0.20 | 0.21 | 0.15 | 0.21 | 0.41 | 0.37 | 0.24 |
| Gutenberg | BERT | ✓ | 0.38 | 0.34 | 0.39 | 0.22 | 0.15 | 0.19 | 0.46 | 0.50 | 0.54 | 0.35 |
| | BERT + Cifar10 | ✓ | 0.55 | 0.55 | 0.55 | 0.49 | 0.49 | 0.40 | 0.50 | 0.55 | 0.51 | 0.51 |
| | | ✗ | 0.45 | 0.46 | 0.35 | 0.36 | 0.34 | 0.16 | 0.32 | 0.32 | 0.34 | 0.34 |
| | BERT + 8 Others | ✓ | 0.50 | 0.59 | 0.60 | 0.50 | 0.43 | 0.43 | 0.45 | 0.60 | 0.57 | 0.52 |
| | | ✗ | 0.56 | 0.57 | 0.52 | 0.42 | 0.36 | 0.24 | 0.24 | 0.34 | 0.31 | 0.40 |
| Isabella | BERT | ✓ | 0.26 | 0.22 | 0.25 | 0.19 | 0.28 | 0.22 | 0.24 | 0.18 | 0.24 | 0.23 |
| | BERT + Cifar10 | ✓ | 0.31 | 0.19 | 0.34 | 0.23 | 0.33 | 0.44 | 0.39 | 0.42 | 0.52 | 0.35 |
| | | ✗ | 0.22 | 0.06 | 0.11 | 0.10 | 0.09 | 0.16 | -0.01 | -0.08 | -0.03 | 0.07 |
| | BERT + 8 Others | ✓ | 0.34 | 0.20 | 0.26 | 0.14 | 0.27 | 0.29 | 0.34 | 0.34 | 0.45 | 0.29 |
| | | ✗ | 0.32 | 0.09 | 0.26 | 0.08 | 0.04 | 0.05 | 0.03 | 0.04 | 0.04 | 0.11 |
| Language | BERT | ✓ | 0.22 | 0.18 | 0.46 | 0.50 | 0.47 | 0.46 | 0.51 | 0.60 | 0.58 | 0.44 |
| | BERT + Cifar10 | ✓ | 0.49 | 0.55 | 0.52 | 0.64 | 0.64 | 0.63 | 0.66 | 0.66 | 0.68 | 0.61 |
| | | ✗ | 0.43 | 0.22 | 0.37 | 0.26 | 0.05 | 0.20 | 0.28 | 0.22 | 0.40 | 0.27 |
| | BERT + 8 Others | ✓ | 0.42 | 0.52 | 0.48 | 0.46 | 0.51 | 0.43 | 0.43 | 0.53 | 0.52 | 0.48 |
| | | ✗ | 0.43 | 0.30 | 0.21 | 0.20 | 0.05 | 0.23 | 0.23 | 0.22 | 0.21 | 0.23 |
| MultNIST | BERT | ✓ | 0.30 | 0.33 | 0.20 | 0.35 | 0.48 | 0.19 | 0.33 | 0.41 | 0.39 | 0.33 |
| | BERT + Cifar10 | ✓ | 0.47 | 0.32 | 0.48 | 0.54 | 0.54 | 0.53 | 0.54 | 0.55 | 0.69 | 0.52 |
| | | ✗ | 0.59 | 0.32 | 0.30 | 0.30 | 0.47 | 0.43 | 0.35 | 0.39 | 0.38 | 0.39 |
| | BERT + 8 Others | ✓ | 0.65 | 0.43 | 0.52 | 0.55 | 0.61 | 0.63 | 0.58 | 0.61 | 0.58 | 0.57 |
| | | ✗ | 0.64 | 0.35 | 0.30 | 0.37 | 0.50 | 0.42 | 0.33 | 0.39 | 0.44 | 0.42 |

Table 12: Results of the ModernBERT-large models in simulated surrogate runs, seed=1

| Task | Model | Train | 100 | 200 | 300 | 400 | 500 | 600 | 700 | 800 | 900 | Avg |
|------|-------|-------|-----|-----|-----|-----|-----|-----|-----|-----|-----|-----|
| AddNIST | BERT | ✓ | 0.19 | 0.36 | 0.13 | 0.16 | 0.24 | 0.10 | 0.12 | 0.39 | 0.31 | 0.22 |
| | BERT + Cifar10 | ✓ | 0.11 | 0.44 | 0.39 | 0.29 | 0.36 | 0.26 | 0.33 | 0.46 | 0.28 | 0.32 |
| | | ✗ | 0.35 | 0.49 | 0.14 | 0.21 | 0.34 | 0.04 | 0.01 | 0.39 | -0.02 | 0.22 |
| | BERT + 8 Others | ✓ | 0.45 | 0.53 | 0.29 | 0.30 | 0.33 | 0.26 | 0.23 | 0.47 | 0.29 | 0.35 |
| | | ✗ | 0.60 | 0.51 | 0.20 | 0.17 | 0.30 | 0.07 | 0.11 | 0.38 | 0.09 | 0.27 |
| Chesseract | BERT | ✓ | 0.36 | 0.24 | 0.36 | 0.22 | 0.39 | 0.28 | 0.44 | 0.45 | 0.51 | 0.36 |
| | BERT + Cifar10 | ✓ | 0.48 | 0.43 | 0.55 | 0.39 | 0.40 | 0.57 | 0.62 | 0.54 | 0.69 | 0.52 |
| | | ✗ | 0.40 | 0.24 | 0.31 | 0.26 | 0.24 | 0.08 | 0.27 | 0.32 | 0.37 | 0.28 |
| | BERT + 8 Others | ✓ | 0.48 | 0.44 | 0.55 | 0.41 | 0.33 | 0.55 | 0.59 | 0.55 | 0.59 | 0.50 |
| | | ✗ | 0.42 | 0.29 | 0.41 | 0.26 | 0.18 | 0.05 | 0.34 | 0.37 | 0.37 | 0.30 |
| CIFARTile | BERT | ✓ | 0.06 | 0.28 | 0.42 | 0.37 | 0.30 | 0.30 | 0.07 | 0.28 | 0.25 | 0.26 |
| | BERT + Cifar10 | ✓ | 0.13 | 0.42 | 0.47 | 0.52 | 0.24 | 0.22 | 0.25 | 0.45 | 0.20 | 0.32 |
| | | ✗ | 0.24 | 0.29 | 0.04 | 0.20 | 0.29 | 0.11 | 0.22 | 0.32 | 0.23 | 0.22 |
| | BERT + 8 Others | ✓ | 0.30 | 0.38 | 0.40 | 0.56 | 0.34 | 0.24 | 0.22 | 0.47 | 0.15 | 0.34 |
| | | ✗ | 0.14 | 0.24 | -0.05 | 0.21 | 0.36 | 0.21 | 0.34 | 0.33 | 0.12 | 0.21 |
| GeoClassing | BERT | ✓ | 0.46 | 0.41 | 0.41 | 0.36 | 0.30 | 0.51 | 0.50 | 0.52 | 0.66 | 0.46 |
| | BERT + Cifar10 | ✓ | 0.57 | 0.55 | 0.59 | 0.49 | 0.49 | 0.55 | 0.56 | 0.57 | 0.65 | 0.56 |
| | | ✗ | 0.56 | 0.49 | 0.51 | 0.27 | 0.13 | 0.15 | 0.39 | 0.25 | 0.31 | 0.34 |
| | BERT + 8 Others | ✓ | 0.66 | 0.58 | 0.57 | 0.43 | 0.51 | 0.60 | 0.56 | 0.61 | 0.70 | 0.58 |
| | | ✗ | 0.62 | 0.54 | 0.45 | 0.10 | 0.08 | 0.08 | 0.23 | 0.23 | 0.22 | 0.28 |
| Gutenberg | BERT | ✓ | 0.17 | 0.56 | 0.59 | 0.56 | 0.49 | 0.49 | 0.41 | 0.54 | 0.50 | 0.48 |
| | BERT + Cifar10 | ✓ | 0.52 | 0.68 | 0.68 | 0.66 | 0.58 | 0.51 | 0.52 | 0.59 | 0.62 | 0.60 |
| | | ✗ | 0.27 | 0.44 | 0.47 | 0.52 | 0.36 | 0.25 | 0.18 | 0.39 | 0.32 | 0.36 |
| | BERT + 8 Others | ✓ | 0.73 | 0.71 | 0.69 | 0.63 | 0.52 | 0.48 | 0.51 | 0.55 | 0.56 | 0.60 |
| | | ✗ | 0.68 | 0.66 | 0.62 | 0.50 | 0.34 | 0.36 | 0.26 | 0.43 | 0.35 | 0.47 |
| Isabella | BERT | ✓ | 0.18 | 0.14 | -0.02 | 0.13 | 0.27 | 0.21 | 0.39 | 0.35 | 0.47 | 0.24 |
| | BERT + Cifar10 | ✓ | 0.34 | 0.27 | 0.29 | 0.42 | 0.59 | 0.62 | 0.62 | 0.66 | 0.63 | 0.49 |
| | | ✗ | 0.26 | 0.05 | -0.03 | -0.14 | -0.06 | 0.06 | -0.01 | 0.05 | 0.14 | 0.04 |
| | BERT + 8 Others | ✓ | 0.30 | 0.14 | 0.27 | 0.44 | 0.63 | 0.69 | 0.64 | 0.63 | 0.66 | 0.49 |
| | | ✗ | 0.27 | 0.05 | 0.02 | 0.05 | 0.11 | 0.16 | 0.10 | 0.12 | 0.16 | 0.12 |
| Language | BERT | ✓ | 0.26 | 0.52 | 0.46 | 0.57 | 0.43 | 0.38 | 0.32 | 0.25 | 0.25 | 0.38 |
| | BERT + Cifar10 | ✓ | 0.49 | 0.60 | 0.61 | 0.66 | 0.62 | 0.56 | 0.26 | 0.31 | 0.44 | 0.51 |
| | | ✗ | 0.41 | 0.38 | 0.41 | 0.33 | 0.35 | 0.44 | 0.21 | 0.29 | 0.39 | 0.36 |
| | BERT + 8 Others | ✓ | 0.33 | 0.56 | 0.53 | 0.70 | 0.47 | 0.49 | 0.30 | 0.35 | 0.38 | 0.46 |
| | | ✗ | 0.34 | 0.47 | 0.43 | 0.38 | 0.40 | 0.46 | 0.25 | 0.24 | 0.27 | 0.36 |
| MultNIST | BERT | ✓ | 0.22 | 0.37 | 0.28 | 0.23 | 0.05 | 0.34 | 0.08 | 0.32 | 0.32 | 0.25 |
| | BERT + Cifar10 | ✓ | 0.41 | 0.46 | 0.42 | 0.72 | 0.40 | 0.56 | 0.54 | 0.54 | 0.47 | 0.50 |
| | | ✗ | 0.38 | 0.32 | 0.39 | 0.59 | 0.38 | 0.48 | 0.26 | 0.39 | 0.40 | 0.40 |
| | BERT + 8 Others | ✓ | 0.52 | 0.44 | 0.40 | 0.61 | 0.31 | 0.52 | 0.19 | 0.53 | 0.55 | 0.45 |
| | | ✗ | 0.56 | 0.39 | 0.35 | 0.53 | 0.35 | 0.45 | 0.34 | 0.43 | 0.39 | 0.42 |

Table 13: Results of the RF Regressor models in simulated surrogate runs, seed=0

| Task | Model | Train | 100 | 200 | 300 | 400 | 500 | 600 | 700 | 800 | 900 | AVG |
|------|-------|-------|-----|-----|-----|-----|-----|-----|-----|-----|-----|-----|
| AddNIST | RF | ✓ | 0.41 | 0.58 | 0.27 | 0.19 | 0.50 | 0.26 | 0.24 | 0.30 | 0.16 | 0.32 |
| | RF + CIFAR10 | ✗ | 0.40 | 0.22 | 0.29 | 0.15 | 0.49 | 0.13 | 0.21 | 0.17 | 0.02 | 0.23 |
| | | ✓ | 0.40 | 0.56 | 0.25 | 0.23 | 0.56 | 0.30 | 0.28 | 0.29 | 0.20 | 0.34 |
| | RF + 8 Others | ✗ | 0.45 | 0.19 | 0.28 | 0.15 | 0.39 | 0.21 | 0.10 | 0.04 | 0.01 | 0.20 |
| | | ✓ | 0.40 | 0.47 | 0.28 | 0.36 | 0.52 | 0.30 | 0.25 | 0.24 | 0.17 | 0.33 |
| Chesseract | RF | ✓ | 0.32 | 0.49 | 0.58 | 0.61 | 0.62 | 0.71 | 0.57 | 0.56 | 0.56 | 0.56 |
| | RF + CIFAR10 | ✗ | 0.37 | 0.42 | 0.44 | 0.40 | 0.32 | 0.46 | 0.33 | 0.39 | 0.34 | 0.39 |
| | | ✓ | 0.42 | 0.47 | 0.43 | 0.61 | 0.62 | 0.70 | 0.59 | 0.54 | 0.54 | 0.55 |
| | RF + 8 Others | ✗ | 0.39 | 0.48 | 0.60 | 0.56 | 0.49 | 0.53 | 0.46 | 0.39 | 0.42 | 0.48 |
| | | ✓ | 0.44 | 0.42 | 0.55 | 0.57 | 0.59 | 0.65 | 0.52 | 0.48 | 0.52 | 0.53 |
| CIFARTile | RF | ✓ | 0.45 | 0.28 | 0.40 | 0.37 | 0.44 | 0.23 | 0.40 | 0.53 | 0.54 | 0.41 |
| | RF + CIFAR10 | ✗ | 0.43 | 0.20 | 0.06 | 0.22 | 0.28 | 0.26 | 0.34 | 0.36 | 0.31 | 0.27 |
| | | ✓ | 0.46 | 0.29 | 0.42 | 0.43 | 0.46 | 0.22 | 0.37 | 0.42 | 0.52 | 0.40 |
| | RF + 8 Others | ✗ | 0.38 | 0.12 | 0.01 | 0.03 | 0.04 | 0.10 | 0.41 | 0.44 | 0.40 | 0.21 |
| | | ✓ | 0.41 | 0.17 | 0.35 | 0.37 | 0.38 | 0.17 | 0.35 | 0.48 | 0.46 | 0.35 |
| GeoClassing | RF | ✓ | 0.17 | 0.06 | 0.17 | 0.51 | 0.51 | 0.51 | 0.66 | 0.75 | 0.57 | 0.44 |
| | RF + CIFAR10 | ✗ | 0.18 | 0.16 | 0.24 | 0.21 | 0.08 | 0.08 | -0.02 | 0.21 | 0.33 | 0.16 |
| | | ✓ | 0.25 | 0.27 | 0.16 | 0.43 | 0.51 | 0.51 | 0.63 | 0.71 | 0.57 | 0.45 |
| | RF + 8 Others | ✗ | 0.29 | 0.37 | 0.19 | 0.25 | 0.18 | 0.12 | 0.13 | 0.22 | 0.26 | 0.22 |
| | | ✓ | 0.30 | 0.32 | 0.04 | 0.41 | 0.57 | 0.56 | 0.64 | 0.70 | 0.62 | 0.46 |
| Gutenberg | RF | ✓ | 0.53 | 0.54 | 0.53 | 0.43 | 0.38 | 0.47 | 0.61 | 0.70 | 0.66 | 0.54 |
| | RF + CIFAR10 | ✗ | 0.50 | 0.56 | 0.51 | 0.42 | 0.25 | 0.23 | 0.42 | 0.37 | 0.44 | 0.41 |
| | | ✓ | 0.56 | 0.59 | 0.53 | 0.41 | 0.29 | 0.39 | 0.57 | 0.69 | 0.63 | 0.52 |
| | RF + 8 Others | ✗ | 0.52 | 0.19 | 0.15 | 0.28 | 0.16 | 0.14 | 0.25 | 0.26 | 0.38 | 0.26 |
| | | ✓ | 0.54 | 0.37 | 0.48 | 0.35 | 0.12 | 0.32 | 0.59 | 0.72 | 0.61 | 0.46 |
| Isabella | RF | ✓ | 0.40 | 0.30 | 0.28 | 0.46 | 0.41 | 0.52 | 0.42 | 0.46 | 0.54 | 0.42 |
| | RF + CIFAR10 | ✗ | 0.37 | 0.13 | 0.09 | 0.13 | 0.17 | 0.32 | 0.01 | 0.03 | 0.16 | 0.16 |
| | | ✓ | 0.40 | 0.36 | 0.25 | 0.31 | 0.32 | 0.43 | 0.37 | 0.46 | 0.43 | 0.37 |
| | RF + 8 Others | ✗ | 0.30 | 0.19 | 0.04 | 0.13 | 0.16 | 0.21 | 0.04 | 0.01 | 0.08 | 0.13 |
| | | ✓ | 0.48 | 0.29 | 0.27 | 0.35 | 0.32 | 0.41 | 0.36 | 0.48 | 0.46 | 0.38 |
| Language | RF | ✓ | 0.26 | 0.49 | 0.52 | 0.60 | 0.72 | 0.57 | 0.60 | 0.61 | 0.67 | 0.56 |
| | RF + CIFAR10 | ✗ | 0.27 | 0.00 | 0.29 | 0.26 | 0.08 | 0.10 | 0.18 | 0.24 | 0.35 | 0.20 |
| | | ✓ | 0.28 | 0.30 | 0.44 | 0.59 | 0.66 | 0.47 | 0.39 | 0.44 | 0.59 | 0.46 |
| | RF + 8 Others | ✗ | 0.35 | 0.17 | 0.32 | 0.18 | 0.07 | 0.12 | 0.21 | 0.24 | 0.29 | 0.22 |
| | | ✓ | 0.36 | 0.30 | 0.41 | 0.38 | 0.35 | 0.34 | 0.47 | 0.47 | 0.53 | 0.40 |
| MultNIST | RF | ✓ | 0.42 | 0.25 | 0.39 | 0.50 | 0.27 | 0.37 | 0.41 | 0.51 | 0.58 | 0.41 |
| | RF + CIFAR10 | ✗ | 0.48 | 0.14 | -0.01 | 0.07 | 0.26 | 0.26 | 0.27 | 0.26 | 0.36 | 0.23 |
| | | ✓ | 0.30 | 0.29 | 0.30 | 0.40 | 0.36 | 0.40 | 0.45 | 0.42 | 0.64 | 0.40 |
| | RF + 8 Others | ✗ | 0.46 | 0.27 | 0.15 | 0.08 | 0.37 | 0.35 | 0.28 | 0.13 | 0.43 | 0.28 |
| | | ✓ | 0.35 | 0.28 | 0.43 | 0.37 | 0.47 | 0.44 | 0.46 | 0.37 | 0.61 | 0.42 |

Table 14: Results of the RF Regressor models in simulated surrogate runs, seed=1

| Task | Model | Train | 100 | 200 | 300 | 400 | 500 | 600 | 700 | 800 | 900 | AVG |
|---|---|---|---|---|---|---|---|---|---|---|---|---|
| AddNIST | RF | ✓ | 0.35 | 0.33 | 0.25 | 0.11 | 0.03 | 0.17 | 0.06 | 0.36 | 0.25 | 0.21 |
| | RF + CIFAR10 | ✗ | 0.13 | 0.52 | 0.11 | 0.21 | 0.35 | 0.25 | 0.30 | 0.23 | 0.26 | 0.26 |
| | | ✓ | 0.41 | 0.42 | 0.24 | 0.20 | 0.01 | 0.15 | 0.19 | 0.33 | 0.28 | 0.25 |
| | RF + 8 Others | ✗ | 0.19 | 0.16 | -0.03 | 0.01 | 0.28 | 0.01 | 0.09 | 0.25 | 0.17 | 0.13 |
| | | ✓ | 0.20 | 0.07 | 0.29 | 0.22 | 0.09 | 0.21 | 0.12 | 0.29 | 0.27 | 0.19 |
| Chesseract | RF | ✓ | 0.50 | 0.48 | 0.46 | 0.53 | 0.48 | 0.52 | 0.60 | 0.64 | 0.66 | 0.54 |
| | RF + CIFAR10 | ✗ | 0.52 | 0.27 | 0.15 | 0.25 | 0.25 | 0.16 | 0.22 | 0.33 | 0.32 | 0.27 |
| | | ✓ | 0.42 | 0.34 | 0.49 | 0.39 | 0.41 | 0.46 | 0.61 | 0.58 | 0.63 | 0.48 |
| | RF + 8 Others | ✗ | 0.49 | 0.24 | 0.30 | 0.24 | 0.25 | 0.26 | 0.26 | 0.14 | 0.41 | 0.29 |
| | | ✓ | 0.52 | 0.42 | 0.45 | 0.34 | 0.37 | 0.49 | 0.52 | 0.55 | 0.59 | 0.47 |
| CIFARTile | RF | ✓ | 0.34 | 0.37 | 0.46 | 0.51 | 0.33 | 0.28 | 0.35 | 0.41 | 0.18 | 0.36 |
| | RF + CIFAR10 | ✗ | 0.31 | 0.34 | 0.28 | 0.41 | 0.37 | 0.35 | 0.33 | 0.23 | 0.26 | 0.32 |
| | | ✓ | 0.37 | 0.50 | 0.43 | 0.58 | 0.32 | 0.32 | 0.38 | 0.37 | 0.29 | 0.40 |
| | RF + 8 Others | ✗ | 0.27 | 0.33 | 0.14 | 0.36 | 0.33 | 0.36 | 0.41 | 0.39 | 0.28 | 0.32 |
| | | ✓ | 0.30 | 0.51 | 0.46 | 0.55 | 0.32 | 0.36 | 0.41 | 0.39 | 0.28 | 0.40 |
| GeoClassing | RF | ✓ | 0.49 | 0.51 | 0.61 | 0.57 | 0.55 | 0.57 | 0.58 | 0.67 | 0.68 | 0.58 |
| | RF + CIFAR10 | ✗ | 0.50 | 0.53 | 0.46 | 0.28 | 0.24 | 0.27 | 0.15 | 0.18 | 0.30 | 0.32 |
| | | ✓ | 0.48 | 0.58 | 0.55 | 0.61 | 0.55 | 0.55 | 0.58 | 0.62 | 0.73 | 0.58 |
| | RF + 8 Others | ✗ | 0.35 | 0.27 | 0.27 | 0.40 | 0.21 | 0.30 | 0.33 | 0.36 | 0.42 | 0.32 |
| | | ✓ | 0.44 | 0.49 | 0.52 | 0.54 | 0.46 | 0.49 | 0.53 | 0.58 | 0.70 | 0.53 |
| Gutenberg | RF | ✓ | 0.52 | 0.60 | 0.62 | 0.60 | 0.54 | 0.39 | 0.54 | 0.65 | 0.66 | 0.57 |
| | RF + CIFAR10 | ✗ | 0.36 | 0.31 | 0.22 | 0.45 | 0.31 | 0.14 | -0.01 | 0.34 | 0.31 | 0.27 |
| | | ✓ | 0.42 | 0.55 | 0.62 | 0.54 | 0.49 | 0.39 | 0.53 | 0.64 | 0.66 | 0.54 |
| | RF + 8 Others | ✗ | 0.39 | 0.44 | 0.37 | 0.47 | 0.30 | 0.16 | 0.26 | 0.45 | 0.32 | 0.35 |
| | | ✓ | 0.42 | 0.50 | 0.61 | 0.51 | 0.48 | 0.33 | 0.44 | 0.57 | 0.57 | 0.49 |
| Isabella | RF | ✓ | 0.34 | 0.33 | 0.27 | 0.66 | 0.67 | 0.69 | 0.69 | 0.70 | 0.57 | 0.55 |
| | RF + CIFAR10 | ✗ | 0.33 | 0.16 | -0.05 | 0.09 | 0.18 | 0.21 | 0.18 | 0.20 | 0.11 | 0.16 |
| | | ✓ | 0.30 | 0.38 | 0.28 | 0.56 | 0.68 | 0.71 | 0.66 | 0.69 | 0.70 | 0.55 |
| | RF + 8 Others | ✗ | 0.35 | 0.14 | 0.10 | 0.12 | 0.19 | 0.27 | 0.27 | 0.32 | 0.19 | 0.22 |
| | | ✓ | 0.32 | 0.34 | 0.30 | 0.61 | 0.68 | 0.72 | 0.66 | 0.65 | 0.67 | 0.55 |
| Language | RF | ✓ | 0.41 | 0.54 | 0.61 | 0.72 | 0.55 | 0.32 | 0.45 | 0.42 | 0.46 | 0.50 |
| | RF + CIFAR10 | ✗ | 0.32 | 0.41 | 0.37 | 0.37 | 0.37 | 0.41 | 0.16 | 0.34 | 0.37 | 0.35 |
| | | ✓ | 0.39 | 0.60 | 0.51 | 0.70 | 0.57 | 0.35 | 0.41 | 0.45 | 0.45 | 0.49 |
| | RF + 8 Others | ✗ | 0.36 | 0.41 | 0.28 | 0.38 | 0.30 | 0.36 | 0.15 | 0.13 | 0.35 | 0.30 |
| | | ✓ | 0.41 | 0.56 | 0.52 | 0.67 | 0.54 | 0.31 | 0.37 | 0.41 | 0.34 | 0.46 |
| MultNIST | RF | ✓ | 0.38 | 0.48 | 0.42 | 0.39 | 0.23 | 0.37 | 0.06 | 0.39 | 0.37 | 0.34 |
| | RF + CIFAR10 | ✗ | 0.32 | 0.21 | 0.21 | 0.43 | 0.26 | 0.37 | 0.24 | 0.31 | 0.25 | 0.29 |
| | | ✓ | 0.50 | 0.43 | 0.39 | 0.53 | 0.22 | 0.31 | 0.11 | 0.34 | 0.41 | 0.36 |
| | RF + 8 Others | ✗ | 0.17 | -0.05 | 0.06 | 0.29 | 0.23 | 0.38 | 0.43 | 0.41 | 0.33 | 0.25 |
| | | ✓ | 0.19 | 0.30 | 0.39 | 0.35 | 0.21 | 0.38 | 0.23 | 0.42 | 0.40 | 0.32 |

Table 15: Results of the XGB Regressor models in simulated surrogate runs, seed=0

| Task | Model | Train | 100 | 200 | 300 | 400 | 500 | 600 | 700 | 800 | 900 | AVG |
|------|-------|-------|-----|-----|-----|-----|-----|-----|-----|-----|-----|-----|
| AddNIST | XGB | ✓ | 0.35 | 0.56 | 0.29 | 0.27 | 0.53 | 0.22 | 0.27 | 0.36 | 0.16 | 0.33 |
| | XGB + CIFAR10 | ✗ | 0.40 | 0.22 | 0.28 | 0.10 | 0.39 | 0.10 | 0.18 | 0.07 | 0.01 | 0.19 |
| | | ✓ | 0.38 | 0.56 | 0.26 | 0.27 | 0.57 | 0.22 | 0.29 | 0.32 | 0.13 | 0.33 |
| | XGB + 8 Others | ✗ | 0.43 | 0.34 | 0.25 | 0.03 | 0.39 | 0.14 | 0.26 | 0.14 | -0.00 | 0.22 |
| | | ✓ | 0.45 | 0.53 | 0.30 | 0.41 | 0.59 | 0.25 | 0.27 | 0.27 | 0.16 | 0.36 |
| Chesseract | XGB | ✓ | 0.38 | 0.47 | 0.60 | 0.53 | 0.60 | 0.67 | 0.55 | 0.53 | 0.55 | 0.54 |
| | XGB + CIFAR10 | ✗ | 0.36 | 0.32 | 0.38 | 0.38 | 0.36 | 0.38 | 0.25 | 0.33 | 0.28 | 0.34 |
| | | ✓ | 0.36 | 0.56 | 0.57 | 0.61 | 0.61 | 0.69 | 0.53 | 0.52 | 0.54 | 0.55 |
| | XGB + 8 Others | ✗ | 0.32 | 0.48 | 0.44 | 0.51 | 0.50 | 0.44 | 0.38 | 0.30 | 0.30 | 0.41 |
| | | ✓ | 0.40 | 0.59 | 0.57 | 0.59 | 0.59 | 0.69 | 0.52 | 0.53 | 0.50 | 0.55 |
| CIFARTile | XGB | ✓ | 0.19 | 0.31 | 0.44 | 0.40 | 0.40 | 0.24 | 0.37 | 0.50 | 0.53 | 0.38 |
| | XGB + CIFAR10 | ✗ | 0.41 | 0.27 | 0.02 | 0.24 | 0.24 | 0.29 | 0.36 | 0.28 | 0.39 | 0.28 |
| | | ✓ | 0.48 | 0.36 | 0.40 | 0.39 | 0.42 | 0.19 | 0.34 | 0.41 | 0.54 | 0.39 |
| | XGB + 8 Others | ✗ | 0.32 | 0.02 | -0.14 | -0.08 | -0.14 | 0.04 | 0.26 | 0.23 | 0.28 | 0.09 |
| | | ✓ | 0.35 | 0.15 | 0.26 | 0.39 | 0.34 | 0.13 | 0.36 | 0.40 | 0.53 | 0.32 |
| GeoClassing | XGB | ✓ | 0.21 | 0.04 | 0.25 | 0.52 | 0.52 | 0.54 | 0.68 | 0.74 | 0.63 | 0.46 |
| | XGB + CIFAR10 | ✗ | 0.09 | -0.05 | 0.14 | 0.12 | 0.01 | 0.00 | -0.16 | -0.03 | 0.32 | 0.05 |
| | | ✓ | 0.25 | 0.08 | 0.15 | 0.52 | 0.58 | 0.56 | 0.70 | 0.79 | 0.68 | 0.48 |
| | XGB + 8 Others | ✗ | 0.28 | 0.42 | -0.00 | 0.22 | 0.14 | 0.15 | 0.15 | 0.27 | 0.35 | 0.22 |
| | | ✓ | 0.29 | 0.32 | -0.00 | 0.51 | 0.59 | 0.52 | 0.69 | 0.75 | 0.66 | 0.48 |
| Gutenberg | XGB | ✓ | 0.43 | 0.44 | 0.50 | 0.42 | 0.30 | 0.41 | 0.59 | 0.71 | 0.65 | 0.49 |
| | XGB + CIFAR10 | ✗ | 0.52 | 0.58 | 0.53 | 0.47 | 0.43 | 0.27 | 0.49 | 0.39 | 0.41 | 0.46 |
| | | ✓ | 0.55 | 0.57 | 0.53 | 0.43 | 0.28 | 0.41 | 0.63 | 0.67 | 0.62 | 0.52 |
| | XGB + 8 Others | ✗ | 0.43 | 0.38 | 0.26 | 0.32 | 0.24 | 0.17 | 0.25 | 0.28 | 0.40 | 0.30 |
| | | ✓ | 0.57 | 0.47 | 0.43 | 0.33 | 0.23 | 0.41 | 0.55 | 0.64 | 0.63 | 0.47 |
| Isabella | XGB | ✓ | 0.40 | 0.27 | 0.31 | 0.40 | 0.38 | 0.52 | 0.41 | 0.48 | 0.51 | 0.41 |
| | XGB + CIFAR10 | ✗ | 0.23 | 0.18 | 0.07 | 0.20 | 0.11 | 0.21 | 0.01 | 0.17 | 0.23 | 0.16 |
| | | ✓ | 0.36 | 0.37 | 0.19 | 0.34 | 0.30 | 0.47 | 0.40 | 0.41 | 0.46 | 0.37 |
| | XGB + 8 Others | ✗ | 0.24 | 0.15 | 0.14 | 0.11 | 0.11 | 0.20 | 0.02 | 0.09 | 0.17 | 0.14 |
| | | ✓ | 0.37 | 0.30 | 0.22 | 0.29 | 0.30 | 0.36 | 0.38 | 0.42 | 0.50 | 0.35 |
| Language | XGB | ✓ | 0.33 | 0.50 | 0.50 | 0.49 | 0.71 | 0.58 | 0.58 | 0.57 | 0.67 | 0.55 |
| | XGB + CIFAR10 | ✗ | 0.35 | 0.06 | 0.29 | 0.25 | 0.07 | 0.15 | 0.18 | 0.26 | 0.34 | 0.22 |
| | | ✓ | 0.31 | 0.38 | 0.42 | 0.56 | 0.61 | 0.48 | 0.49 | 0.59 | 0.65 | 0.50 |
| | XGB + 8 Others | ✗ | 0.40 | 0.21 | 0.36 | 0.25 | 0.09 | 0.12 | 0.23 | 0.29 | 0.31 | 0.25 |
| | | ✓ | 0.38 | 0.36 | 0.50 | 0.46 | 0.42 | 0.43 | 0.45 | 0.55 | 0.62 | 0.46 |
| MultNIST | XGB | ✓ | 0.30 | 0.24 | 0.46 | 0.50 | 0.36 | 0.39 | 0.44 | 0.52 | 0.57 | 0.42 |
| | XGB + CIFAR10 | ✗ | 0.44 | 0.16 | -0.05 | -0.02 | 0.19 | 0.35 | 0.35 | 0.25 | 0.43 | 0.23 |
| | | ✓ | 0.23 | 0.28 | 0.44 | 0.47 | 0.36 | 0.41 | 0.51 | 0.46 | 0.60 | 0.42 |
| | XGB + 8 Others | ✗ | 0.50 | 0.25 | 0.22 | 0.00 | 0.42 | 0.37 | 0.33 | 0.21 | 0.49 | 0.31 |
| | | ✓ | 0.39 | 0.21 | 0.40 | 0.40 | 0.51 | 0.44 | 0.47 | 0.42 | 0.60 | 0.43 |

Table 16: Results of the XGB Regressor models in simulated surrogate runs, seed=1

| Task | Model | Train | 100 | 200 | 300 | 400 | 500 | 600 | 700 | 800 | 900 | AVG |
|---|---|---|---|---|---|---|---|---|---|---|---|---|
| AddNIST | XGB | ✓ | 0.34 | 0.42 | 0.24 | 0.11 | -0.02 | 0.26 | 0.23 | 0.37 | 0.26 | 0.24 |
| | XGB + CIFAR10 | ✗ | -0.13 | 0.50 | 0.11 | 0.22 | 0.35 | 0.26 | 0.32 | 0.26 | 0.24 | 0.24 |
| | | ✓ | 0.37 | 0.45 | 0.27 | 0.21 | 0.09 | 0.25 | 0.34 | 0.37 | 0.27 | 0.29 |
| | XGB + 8 Others | ✗ | 0.26 | 0.29 | 0.16 | 0.28 | 0.39 | 0.28 | 0.22 | 0.29 | 0.22 | 0.26 |
| | | ✓ | 0.24 | 0.36 | 0.28 | 0.21 | 0.19 | 0.22 | 0.31 | 0.38 | 0.29 | 0.28 |
| Chesseract | XGB | ✓ | 0.52 | 0.51 | 0.50 | 0.50 | 0.48 | 0.52 | 0.60 | 0.60 | 0.65 | 0.54 |
| | XGB + CIFAR10 | ✗ | 0.50 | 0.33 | 0.17 | 0.17 | 0.25 | 0.21 | 0.27 | 0.36 | 0.35 | 0.29 |
| | | ✓ | 0.41 | 0.38 | 0.49 | 0.39 | 0.42 | 0.50 | 0.59 | 0.59 | 0.60 | 0.48 |
| | XGB + 8 Others | ✗ | 0.54 | 0.29 | 0.31 | 0.17 | 0.22 | 0.35 | 0.31 | 0.35 | 0.52 | 0.34 |
| | | ✓ | 0.54 | 0.43 | 0.52 | 0.37 | 0.44 | 0.50 | 0.44 | 0.57 | 0.56 | 0.49 |
| CIFARTile | XGB | ✓ | 0.29 | 0.34 | 0.46 | 0.50 | 0.30 | 0.27 | 0.31 | 0.43 | 0.16 | 0.34 |
| | XGB + CIFAR10 | ✗ | 0.22 | 0.31 | 0.11 | 0.34 | 0.30 | 0.21 | 0.28 | 0.25 | 0.37 | 0.27 |
| | | ✓ | 0.29 | 0.52 | 0.46 | 0.54 | 0.33 | 0.28 | 0.38 | 0.47 | 0.25 | 0.39 |
| | XGB + 8 Others | ✗ | 0.34 | 0.33 | 0.27 | 0.41 | 0.27 | 0.19 | 0.32 | 0.29 | 0.30 | 0.30 |
| | | ✓ | 0.34 | 0.47 | 0.46 | 0.57 | 0.29 | 0.28 | 0.40 | 0.45 | 0.25 | 0.39 |
| GeoClassing | XGB | ✓ | 0.51 | 0.51 | 0.54 | 0.59 | 0.57 | 0.57 | 0.60 | 0.62 | 0.70 | 0.58 |
| | XGB + CIFAR10 | ✗ | 0.52 | 0.45 | 0.40 | 0.37 | 0.23 | 0.27 | 0.27 | 0.22 | 0.32 | 0.34 |
| | | ✓ | 0.50 | 0.62 | 0.56 | 0.60 | 0.57 | 0.60 | 0.63 | 0.67 | 0.72 | 0.61 |
| | XGB + 8 Others | ✗ | 0.33 | 0.34 | 0.32 | 0.35 | 0.24 | 0.37 | 0.36 | 0.31 | 0.33 | 0.33 |
| | | ✓ | 0.40 | 0.54 | 0.53 | 0.52 | 0.52 | 0.59 | 0.57 | 0.64 | 0.72 | 0.56 |
| Gutenberg | XGB | ✓ | 0.56 | 0.63 | 0.65 | 0.57 | 0.56 | 0.40 | 0.54 | 0.65 | 0.65 | 0.58 |
| | XGB + CIFAR10 | ✗ | 0.30 | 0.35 | 0.16 | 0.44 | 0.29 | 0.03 | 0.01 | 0.33 | 0.35 | 0.25 |
| | | ✓ | 0.37 | 0.63 | 0.66 | 0.56 | 0.56 | 0.42 | 0.55 | 0.62 | 0.60 | 0.55 |
| | XGB + 8 Others | ✗ | 0.39 | 0.54 | 0.49 | 0.51 | 0.23 | 0.21 | 0.24 | 0.31 | 0.31 | 0.36 |
| | | ✓ | 0.43 | 0.64 | 0.67 | 0.58 | 0.52 | 0.40 | 0.53 | 0.57 | 0.60 | 0.55 |
| Isabella | XGB | ✓ | 0.25 | 0.32 | 0.42 | 0.69 | 0.68 | 0.70 | 0.69 | 0.71 | 0.60 | 0.56 |
| | XGB + CIFAR10 | ✗ | 0.33 | 0.24 | -0.01 | 0.02 | 0.07 | 0.09 | 0.05 | 0.18 | 0.08 | 0.12 |
| | | ✓ | 0.30 | 0.41 | 0.38 | 0.63 | 0.70 | 0.72 | 0.65 | 0.71 | 0.68 | 0.58 |
| | XGB + 8 Others | ✗ | 0.32 | 0.29 | 0.14 | 0.08 | 0.08 | 0.12 | 0.11 | 0.25 | 0.14 | 0.17 |
| | | ✓ | 0.31 | 0.38 | 0.36 | 0.67 | 0.73 | 0.72 | 0.65 | 0.68 | 0.66 | 0.57 |
| Language | XGB | ✓ | 0.47 | 0.54 | 0.56 | 0.70 | 0.57 | 0.32 | 0.33 | 0.36 | 0.53 | 0.49 |
| | XGB + CIFAR10 | ✗ | 0.41 | 0.36 | 0.37 | 0.37 | 0.39 | 0.39 | 0.16 | 0.33 | 0.38 | 0.35 |
| | | ✓ | 0.37 | 0.56 | 0.60 | 0.64 | 0.60 | 0.35 | 0.33 | 0.34 | 0.42 | 0.47 |
| | XGB + 8 Others | ✗ | 0.40 | 0.41 | 0.46 | 0.41 | 0.36 | 0.35 | 0.18 | 0.28 | 0.35 | 0.36 |
| | | ✓ | 0.46 | 0.55 | 0.56 | 0.65 | 0.52 | 0.31 | 0.40 | 0.36 | 0.36 | 0.46 |
| MultNIST | XGB | ✓ | 0.44 | 0.46 | 0.45 | 0.42 | 0.20 | 0.47 | 0.12 | 0.41 | 0.41 | 0.37 |
| | XGB + CIFAR10 | ✗ | 0.10 | -0.05 | -0.03 | 0.36 | 0.21 | 0.35 | 0.26 | 0.38 | 0.32 | 0.21 |
| | | ✓ | 0.35 | 0.46 | 0.39 | 0.37 | 0.26 | 0.42 | 0.13 | 0.42 | 0.42 | 0.36 |
| | XGB + 8 Others | ✗ | 0.06 | -0.02 | 0.04 | 0.42 | 0.22 | 0.40 | 0.32 | 0.35 | 0.21 | 0.22 |
| | | ✓ | 0.07 | 0.33 | 0.38 | 0.30 | 0.21 | 0.46 | 0.17 | 0.45 | 0.43 | 0.31 |

Table 17: Comparison of RF and XGBoost without transfer. Average Kendall tau over all Unseen datasets and seeds when trained on data from first iterations. Only the given dataset is used for training.

| | RF | | | | | | | | |
|---|---|---|---|---|---|---|---|---|---|
| Remove zero acc. | limit Aggregation | 100 | 200 | 300 | 400 | 500 | 600 | 700 | 800 | 900 |
| False | minmax | 0.39 | 0.42 | 0.44 | 0.50 | 0.46 | 0.44 | 0.47 | 0.54 | 0.49 |
| | none | 0.39 | 0.42 | 0.44 | 0.50 | 0.46 | 0.44 | 0.47 | 0.54 | 0.49 |
| | percentile | 0.37 | 0.42 | 0.44 | 0.51 | 0.46 | 0.45 | 0.46 | 0.54 | 0.51 |
| True | minmax | 0.39 | 0.41 | 0.43 | 0.49 | 0.45 | 0.43 | 0.45 | 0.54 | 0.51 |
| | none | 0.39 | 0.41 | 0.43 | 0.49 | 0.45 | 0.43 | 0.45 | 0.54 | 0.51 |
| | percentile | 0.37 | 0.41 | 0.42 | 0.50 | 0.46 | 0.45 | 0.46 | 0.54 | 0.50 |

| | XGB | | | | | | | | |
|---|---|---|---|---|---|---|---|---|---|
| Remove zero acc. | limit Aggregation | 100 | 200 | 300 | 400 | 500 | 600 | 700 | 800 | 900 |
| False | minmax | 0.37 | 0.41 | 0.44 | 0.48 | 0.47 | 0.44 | 0.46 | 0.52 | 0.50 |
| | none | 0.37 | 0.41 | 0.44 | 0.48 | 0.47 | 0.44 | 0.46 | 0.52 | 0.50 |
| | percentile | 0.35 | 0.40 | 0.44 | 0.50 | 0.47 | 0.45 | 0.48 | 0.55 | 0.52 |
| True | minmax | 0.37 | 0.41 | 0.45 | 0.48 | 0.45 | 0.44 | 0.46 | 0.54 | 0.51 |
| | none | 0.37 | 0.41 | 0.45 | 0.48 | 0.45 | 0.44 | 0.46 | 0.54 | 0.51 |
| | percentile | 0.36 | 0.42 | 0.43 | 0.49 | 0.45 | 0.46 | 0.47 | 0.55 | 0.51 |

Table 18: Comparison of RF and XGBoost with transfer. Average Kendall tau over all Unseen datasets and seeds when trained on data from first iterations. Data from the initial part of the current dataset and Cifar10, the other Unseen datasets, or both are used for training.

| | RF | | | | | | | | |
|---|---|---|---|---|---|---|---|---|---|
| Remove zero acc. | limit Aggregation | 100 | 200 | 300 | 400 | 500 | 600 | 700 | 800 | 900 |
| False | minmax | 0.28 | 0.31 | 0.28 | 0.31 | 0.33 | 0.31 | 0.33 | 0.36 | 0.37 |
| | none | 0.31 | 0.30 | 0.27 | 0.32 | 0.33 | 0.31 | 0.31 | 0.35 | 0.35 |
| | percentile | 0.35 | 0.33 | 0.31 | 0.36 | 0.36 | 0.35 | 0.37 | 0.40 | 0.41 |
| True | minmax | 0.37 | 0.33 | 0.31 | 0.35 | 0.34 | 0.33 | 0.33 | 0.37 | 0.39 |
| | none | 0.33 | 0.29 | 0.26 | 0.31 | 0.33 | 0.30 | 0.29 | 0.34 | 0.35 |
| | percentile | 0.36 | 0.32 | 0.31 | 0.35 | 0.35 | 0.34 | 0.35 | 0.40 | 0.41 |

| | XGB | | | | | | | | |
|---|---|---|---|---|---|---|---|---|---|
| Remove zero acc. | limit Aggregation | 100 | 200 | 300 | 400 | 500 | 600 | 700 | 800 | 900 |
| False | minmax | 0.29 | 0.31 | 0.29 | 0.32 | 0.32 | 0.31 | 0.33 | 0.35 | 0.35 |
| | none | 0.29 | 0.31 | 0.26 | 0.31 | 0.31 | 0.30 | 0.31 | 0.33 | 0.33 |
| | percentile | 0.32 | 0.34 | 0.31 | 0.36 | 0.37 | 0.35 | 0.36 | 0.40 | 0.40 |
| True | minmax | 0.35 | 0.35 | 0.31 | 0.35 | 0.34 | 0.34 | 0.35 | 0.38 | 0.40 |
| | none | 0.32 | 0.32 | 0.27 | 0.32 | 0.32 | 0.31 | 0.31 | 0.34 | 0.36 |
| | percentile | 0.34 | 0.35 | 0.31 | 0.36 | 0.37 | 0.35 | 0.36 | 0.41 | 0.40 |

## E Prompt

An example prompt for CIFAR10 few-shot learning is provided below. We followed the PP Prompt format (Jawahar et al., 2024) and changed the component name hyperparameters to definitions because `einspace` requires more explanation in this section.

---

**Role**

You are a performance estimator for image classification task, where you will estimate the accuracy for the test architecture. Please output the accuracy directly without anything else.

---

**Instruction**

You should follow these instructions: 1. You should understand that the image classification task is CIFAR10 and the quality of a configuration is measured based on validation accuracy. 2. The CIFAR-10 dataset consists of 60000 32x32 colour images in 10 classes, with 6000 images per class. There are 40000 training images, 10000 validation images and 10000 test images. The 10 classes are airplane, automobile, bird, cat, deer, dog, frog, horse, ship and truck. 3. You should concentrate on the example configurations provided below along with their accuracies to understand the complex relationships between architecture configuration and accuracies.

---

**Definition**

Search Space Definition: The search space includes groups of operations which can represent many state-of-the-art neural architectures. The search space is based on context free grammar and each candidate represents a syntax tree of the architecture.

The four fundamental operations are: 1. Branching: One-to-many functions that direct the flow of information through the network by cloning or splitting tensors. Examples include the branching within self-attention modules into queries, keys and values. 2. Aggregation: Many-to-one functions that merge the information from multiple tensors into one. Examples include matrix multiplication, summation and concatenation. 3. Routing: One-to-one functions that change the shape or the order of the content in a tensor without altering its information. Examples include axis permutations as well as the im2col and col2im operations. 4. Computation: One-to-one functions that alter the information of the tensor, either by parametrised operations, normalisation or non-linearities. Examples include linear layers, batchnorm and activations like ReLU and softmax.

The two feature modes are: 1. Im mode: Maintains a 3D tensor of shape (C, H, W), where C is the number of channels, H is the height and W is the width. Most convolutional architectures operate in this mode. 2. Col mode: Maintains a 2D tensor of shape(S, D),where S is the sequence length and D is the token dimensionality. This is the mode in which most transformer architectures operate.

For each candidate in the search space, its format is described using functions formatted as below: 1. Branching functions: branching(b)[M] - where b is the number of splits/clones, M is a set of other operations. clone(b) - cloning b copies of the tensor. group(b,dim) - splitting tensor into b parts along dimension dim. 2. Aggregation functions: dot_product(scaled) - matrix multiplication with optional scaling add - summation of multiple tensors. concat(b,dim) - concatenate b tensors along dimension d. 3. Routing functions: routing[M] - where M is a set of other operations. im2col(k,s,p) - convert from im mode to col mode, where k is kernel size, s is the stride and p the padding. col2im - convert from col mode to im mode. permute(o) - same as permute function in pytorch. identity - keep original tensor. 4. Computation functions: computation<o> - where o could be any functions listed below. linear(d) - linear layers with d as the output dimension. norm - batch-norm functionality in the Im mode and layer-norm in Col mode. softmax - softmax operation applied to the final dimension. relu - leaky relu activation function. pos-enc -positional encoding.

An example representation of a traditional convolutional block with a skip connection: ...

