# OpenReview forum: "Transferrable Surrogates in Expressive Neural Architecture Search Spaces"
_automl.cc/AutoML/2025/Methods_Track — AutoML 2025 Methods Track_

### Official Review · Reviewer_NZNJ · 2025-04-22

**Comments To Authors:**

Summary

This paper explores the use of surrogate models—especially LLMs and graph-based regressors—for NAS in expressive, grammar-based search spaces like einspace. The authors evaluate various performance predictors across multiple datasets, demonstrating that surrogate-based search significantly accelerates NAS while maintaining or improving performance. They also show that LLM-based surrogates generalize across tasks and can even be used as direct search objectives.

Strengths

- Contribution: The paper targets a growing concern in NAS—scaling search in large, flexible spaces. Addressing this with transferable surrogate models is both practical and novel.


- Strong empirical results: Across 9 datasets (including CIFAR-10 and UnseenNAS), surrogate-guided search, particularly with ModernBERT, consistently outperforms baseline evolutionary search, with speed-ups.


- Evaluation: The authors rigorously evaluate their approach under IID, OOD, and leave-one-out settings with rank correlation metrics and accuracy curves across iterations.


- Well-structured related work: The paper appropriately contextualizes surrogate modeling and LLMs within NAS.

Suggestions for Improvement

- Include or at least discuss multi-objective extensions and potential for hardware-aware NAS, as these are natural use cases for energy-aware surrogate-driven search.


- Investigate why certain datasets lead to poor transfer performance (Isabella, Chesseract)  and whether task-specific surrogate adaptation could improve results.

- Adjust the font sizes of most of the Tables and Figures to improve readability.

This paper offers a promising approach to scalable NAS in expressive search spaces using surrogate models, including LLMs. The method is well-motivated.

**Review Confidence:**

3

**Review Rating:**

7

---

### Official Review · Reviewer_pEkX · 2025-04-30

**Comments To Authors:**

This paepr studies how to build transferable accuracy predictors for large, grammar-based NAS spaces. They propose a tabular and LLM style surrogate and find that LLM surrogate model gives very high rank correlation (however, prior zero-cost proxy research have shown similar rank correlations). Further, their approach demonstrates transferrability and these surrogates can be used to speed up search significantly. It is interesting to demonstrate LLMs can do this task, however, it would be nice to see direct comparison to cross domain predictors [MultiPredict, HELP, FLAN, GENNAPE].

Overall, the paper is well motivated and it is interesting to see LLM surrogates work, but it would be useful to contextualize the results with respect to other NAS papers that focus on generalizable NAS.  My main problem is standardized evaluation, [CATE] also uses transformers for architecture encoding, which seems to be missing. Also, why not use NASBench-101/201 just for baseline numers that contextualize the papers contributions, beyond the grammar-based search space proposed?

[GENNAPE] Mills, Keith G., et al. ‘GENNAPE: Towards Generalized Neural Architecture Performance Estimators’. Proceedings of the AAAI Conference on Artificial Intelligence, vol. 37, no. 8, Association for the Advancement of Artificial Intelligence (AAAI), June 2023, pp. 9190–9199, https://doi.org/10.1609/aaai.v37i8.26102.

[CATE] Yan, Shen, et al. ‘CATE: Computation-Aware Neural Architecture Encoding with Transformers’. arXiv [Cs.LG], 2021, http://arxiv.org/abs/2102.07108. arXiv.

[FLAN] Akhauri, Yash, and Mohamed S. Abdelfattah. ‘Encodings for Prediction-Based Neural Architecture Search’. arXiv [Cs.LG], 2024, http://arxiv.org/abs/2403.02484. arXiv.

[HELP] Lee, Hayeon, et al. ‘HELP: Hardware-Adaptive Efficient Latency Prediction for NAS via Meta-Learning’. arXiv [Cs.LG], 2021, http://arxiv.org/abs/2106.08630. arXiv.

[MultiPredict] Akhauri, Yash, and Mohamed S. Abdelfattah. ‘Multi-Predict: Few Shot Predictors For Efficient Neural Architecture Search’. arXiv [Cs.LG], 2023, http://arxiv.org/abs/2306.02459. arXiv.

**Review Confidence:**

5

**Review Rating:**

4

---

### Official Review · Reviewer_6bPJ · 2025-05-06

**Comments To Authors:**

The paper presents a prediction approach for NAS based on surrogate models and LLMs. The primary contribution of the paper is a large-scale empirical comparison of different approaches.

The paper is well written and investigates an interesting problem. It combines existing methods, but where it really adds something to the literature is the large-scale comparison of different approaches for surrogate models in NAS, in particular LLM-based and non-LLM-based approaches. The authors also evaluate to what extent learned surrogate models for predicting the performance of neural architectures transfer to different tasks.

Altogether, I think that this paper should be accepted as the empirical results are likely to be of interest to a large part of the community.

**Review Confidence:**

3

**Review Rating:**

7

---

### Meta-Review · Area_Chair_RSw8 · 2025-05-07

**Recommendation:** Accept
**Confidence:** 4

**Metareview:**

The submission proposes an empirical study on the use of surrogate models for predicting the accuracy of neural architectures in grammar-based NAS spaces. The paper received two weak acceptances and a weak rejection from the reviewers. The major point of disagreement was the extent to which the prior work and the broader NAS literature were included.

Despite the constructive criticism of reviewer pEkX, I believe the paper presents a substantial empirical study with practical relevance, which is clearly documented. While there is limited novelty in terms of algorithmic development, there are strengths in terms of empirical validation. The positive reproducibility review further highlights the quality of the paper.

As a result, I recommend accepting the paper.

I invite the authors to consider the points raised by reviewer pEkX for the camera-ready version.